# MetaMD: Principled Optimiser Meta-Learning for Deep Learning

## Abstract

Optimiser design influences learning speed and generalisation in training machine learning models. Several studies have attempted to learn more effective gradient-descent optimisers via solving a bi-level optimisation problem where generalisation error is minimised with respect to optimiser parameters. However, most existing neural network oriented optimiser learning methods are intuitively motivated, without clear theoretical support, and focus on learning implicit biases that improve generalisation, rather than speed of convergence. We take a different perspective starting from mirror descent rather than gradient descent, and meta-learning the corresponding Bregman divergence. Within this paradigm, we formalise a novel meta-learning objective of optimising the rate of convergence. The resulting framework, termed Meta Mirror Descent (MetaMD), learns to accelerate optimisation speed. Unlike many meta-learned neural network optimisers, it also supports convergence guarantees and uniquely does so without requiring validation data. We empirically evaluate our framework on a variety of tasks and architectures in terms of convergence rate and generalisation error and demonstrate strong performance.

## 1 Introduction

Gradient-based optimisation algorithms, such as stochastic gradient descent (SGD), are fundamental building blocks of many machine learning algorithms—notably those focused on training linear models and deep neural networks. These methods are typically developed to solve a broad class of problems, and therefore the method developers make as few assumptions about the target problem as possible. This leads to a variety of general purpose techniques for optimisation, but such generality often comes with slower convergence. By taking advantage of more information about the target problem, one is typically able to design more efficient—but less general—optimisation algorithms. Another challenge in a non-convex deep learning context, is that that many of the empirically fastest optimisers such as Adam (Kingma & Ba, 2015) lack convergence guarantees.

While one line of research hand-designs optimisers to exploit known properties of particular problems, a complementary line of research focuses on situations where optimisation problems come in families. This allows the use of meta-learning techniques to fit an optimiser to the given problem family with the goal of maximising convergence speed or generalisation performance. For example, in the many-shot regime, Andrychowicz et al. (2016) and Wichrowska et al. (2017) learn black-box neural optimisers to accelerate training of neural networks, while Bello et al. (2017) learn symbolic gradient-based optimisers to improve generalisation. MAML (Finn et al., 2017) and Meta-SGD (Li et al., 2017) learned initialisation and learning rate for SGD training of neural networks with good generalisation performance in the few-shot regime. Later generalisations focused on learning problem family-specific curvature information (Park & Oliva, 2019; Flennerhag et al., 2020). Nevertheless, most existing learned optimisers such as Andrychowicz et al. (2016); Wichrowska et al. (2017); Flennerhag et al. (2020); Bello et al. (2017) can not provide convergence guarantees.

In this work, we revisit the optimiser learning problem from the perspective of *mirror descent*. Mirror descent introduces a Bregman divergence that regularises the distance between current and next iterate, introducing a strongly convex sub-problem that can be optimised exactly. In mirror descent, the choice of Bregman divergence determines optimisation dynamics. In a meta-learning context, the Bregman divergence thus provides a interesting representation of an optimisation strategy that can be fit to a given family of optimisation problems, leading to our learned optimiser termed Meta Mirror Descent (MetaMD). Many existing learned optimisers do not have a formal notion of convergence

rate, and in practice typically optimise a meta-objective reflecting training or validation loss after a fixed number of iterations. In contrast, MetaMD is directly trained to optimise the convergence rate bound for mirror descent. Importantly, this means we can adapt theoretical guarantees from mirror descent to provide convergence guarantees for MetaMD, an important property not provided by most learned optimsers, and many hand-designed optimisers widely used in deep learning.

An important issue in meta-learning mirror descent is specifying the family of Bregman divergences to learn. Meta-learning with general Bregman divergences leads to an intractable tri-level optimisation problem. Thus, we seek a family of divergences for which the innermost optimisation has a closed form solution. The chosen paramaterisation should be complex enough to exhibit interesting optimisation dynamics, simple enough to provide a closed form solution, and always provide a valid Bregman divergence. We provide a parameterisation that meets all these desideratum in the form of a Kronecker factorised preconditioner Zehfuss (1858); Martens & Grosse (2015). Preconditioners have been successfully exploited in meta-learning methods such as Meta-Curvature Park & Oliva (2019), and WarpGrad Flennerhag et al. (2020), and are richer than the parameter-wise learning rates Li et al. (2017); Khodak et al. (2019), which is a special case. Importantly, we also show how to train the divergence efficiently with implicit gradient computation Lorraine et al. (2020). Uniquely, this means that our framework can effectively train learning-rates (as a special case of preconditioning) with implicit gradient, an idea that was previously suggested to be impossible (Lorraine et al., 2020).

Empirically we demonstrate that we can train MetaMD given a model architecture and a suite of training tasks. We then deploy it to novel testing tasks and show that MetaMD provides fast convergence and good generalisation compared to several existing hand-designed and learned optimisers.

## 2 RELATED WORK

Meta-learning aims to extract some notion of '*how to learn*' given a task or distribution of tasks (Hospedales et al., 2020), such that new learning trials are better or faster. These two stages are often called meta-training, and meta-testing respectively. Key dichotomies include: meta-learning from a single task vs a task distribution (as formalised, e.g., by Baxter (2000)); the type of meta-knowledge to be extracted; and long- vs short-horizon meta-learning. For few-shot problems with short optimisation horizons, the seminal model-agnostic meta-learning (MAML) Finn et al. (2017) learns an initial condition from which only a few optimisation steps are required solve a new task. Meta-SGD Li et al. (2017) and Meta-Curvature Park & Oliva (2019) extend MAML by learning a parameter-wise learning rate, and a preconditioning matrix, respectively. Another group of methods focus on larger scale problems in terms of dataset size and optimisation horizon. For example, neural architecture search (NAS) Real et al. (2019); Zoph & Le (2017) discovers effective neural architectures. MetaReg Balaji et al. (2018) meta-learns regularisation parameters to improve domain generalisation.

Several studies focus specifically on optimiser meta-learning for many-shot batch learning problems, which we address here. In this case, the extracted meta-knowledge spans learning rates for SGD (Micaelli & Storkey, 2021), symbolic gradient-descent rules (Bello et al., 2017) neural network gradient-descent rules (Andrychowicz et al., 2016; Li & Malik, 2017), and gradient-free optimisers (Sandler et al., 2021; Chen et al., 2017). Differently to the gradient-descent based methods, we start from the perspective of mirror descent, where mirror descent's Bregman Divergence provides a target for meta-learning. This perspective has several benefits, notably the ability to derive a learned optimiser with convergence and generalisation guarantees. Our suggested practical instantiation is to meta-learn a Kronecker-factorised pre-conditioner, which is effective and efficient. However, the framework is general and can be used to learn particular special cases such as element-wise learning rates Li et al. (2017); Micaelli & Storkey (2021). Beyond supporting this richer representation, we provide convergence guarantees, do not rely on a validation set, and demonstrate cross-dataset generalisation, enabling us to amortise meta-learning cost. In contrast, the single task meta-learner of Micaelli & Storkey (2021) needs to repeat meta-learning on each specific dataset to optimise per-dataset validation performance.

In addition to the work on deep meta-learning of optimisers and optimiser hyperparameters, there is also a rich literature on principled development of online meta-learning algorithms that aim minimise various notions of meta-regret Finn et al. (2019); Denevi et al. (2019b;a; 2018); Balcan et al. (2019); Khodak et al. (2019). Several of these build on online mirror descent, but there is an important distinction between their work and ours: we focus on the batch setting, with a view to ensuring the resulting algorithm can be tractably applied to deep neural networks. ARUBA (Khodak et al., 2019)

is perhaps closest to our work. A closed-form update rule for the Bregman divergence is derived from a similar convergence bound. However, (i) it addresses online rather than the batch training regime relevant for neural network learning, and (ii) it is only instantiated with a Bregman divergence parameterised by a diagonal matrix.

## 3   STOCHASTIC MIRROR DESCENT

We formalise the problem of learning an optimiser using the Stochastic Mirror Descent (SMD) framework, which can be thought of as a generalisation of stochastic subgradient descent. SMD optimisers produce a series of progressively better estimates for the optimal parameters of the objective function. This is accomplished by solving a convex optimisation problem at each step, $t$,

$$\theta_{t+1} = \arg\min_{\theta} \langle \nabla_{\theta} \mathcal{L}_S(\theta_t; \xi), \theta \rangle + \frac{1}{2\eta} B_{\phi}(\theta || \theta_t) \tag{1}$$

where $\mathcal{L}_S(\cdot; \xi)$ provides an unbiased stochastic approximation of the loss based on the random variable $\xi$ and training set $S$, $\eta$ is the step size. $B_{\phi}$ denotes a Bregman divergence, which can be thought of as a way of measuring distance in parameter space, and each choice of Bregman divergence leads to a different optimisation algorithm. One can define Bregman divergences as

$$B_{\phi}(\theta || \theta') = \phi(\theta) - \phi(\theta') - \langle \nabla \phi(\theta'), \theta - \theta' \rangle, \tag{2}$$

where $\phi$ is a $\lambda$-strongly convex function. There are several choices of $\phi$ that result in existing algorithms specialised for various types of optimisation problems in machine learning. For example, if one chooses $\phi$ to be $\frac{1}{2} || \cdot ||_2^2$, then mirror descent becomes gradient descent, while choosing $\phi$ to be the negative entropy results in the Kullback-Leibler divergence, leading to the exponentiated gradient algorithm (Kivinen & Warmuth, 1997). A significant benefit of deriving new algorithms that fit into the mirror descent framework is that, in the convex setting, one can obtain bounds on the rate of convergence towards a minima $\theta_*$ for any valid choice of $\phi$—including those that are learned. In this work we make use of the following convergence analysis:

**Theorem 3.1** (Corollary 8 from D'Orazio et al. (2021)). *If $\mathcal{L}_S(\cdot; \xi)$ is convex and $L$-smooth[1] w.r.t. to the Euclidean norm for all $\xi$, and $\phi$ is essentially smooth, then stochastic mirror descent with constant step size $\eta \leq \lambda/2L$ guarantees*

$$\mathbb{E}_{\xi}[\mathcal{L}_S(\bar{\theta}; \xi) - \mathcal{L}_S(\theta_*; \xi)] \leq \frac{2B_{\phi}(\theta_* || \theta_1)}{\eta T} + 2\sigma^2, \tag{3}$$

*where $\bar{\theta} = \frac{1}{T} \sum_{t=1}^{T} \theta_t$ and $\sigma^2 = \inf_{\theta} \mathbb{E}_{\xi}[\mathcal{L}_S(\theta; \xi)] - \mathbb{E}_{\xi}[\inf_{\theta} \mathcal{L}_S(\theta; \xi)]$.*

We note that one can obtain a trivial corollary where instead of $\bar{\theta}$ appearing the left-hand side, one can instead use the best iterate obtained after $T$ steps, $\arg\min_{\theta_t : 1 \leq t \leq T} \mathbb{E}_{\xi}[\mathcal{L}_S(\theta_t; \xi)]$. The above bound assumes convexity, but the convergence of stochastic mirror descent for neural networks is an open problem, with recent work focusing on narrower families of optimisers (e.g., Zhang et al. (2020)).

## 4   META-LEARNING A BREGMAN DIVERGENCE

### 4.1   OPTIMISER LEARNING FRAMEWORK

We propose a meta-learning algorithm to learn mirror descent optimisers. We consider the multi-task meta-learning setting (Hospedales et al., 2020; Finn et al., 2017), assuming that a task distribution $p(\mathcal{T})$ is available from which we can draw tasks for meta-training, and that we will evaluate the learned optimizer by meta-testing on novel tasks from the same distribution. For gradient-based meta-learning, the meta-training procedure is conventionally framed as a bilevel optimization problem where the inner problem solves learning tasks given the optimiser, and the outer problem updates the optimiser (Hospedales et al., 2020). We do this by using a parameterised potential function, $\phi_M$, to construct the Bregman divergence. The outer problem is to minimise some meta objective denoted $\mathcal{E}(M)$ with respect to parameters, $M$. Since our mirror descent optimizer is defined by a Bregman

---

[1]In D'Orazio et al. (2021), a function, $f$, is said to be L-smooth if $\frac{1}{2} || \nabla f(\vec{x}) ||_2^2 \leq L$ for all $\vec{x}$.

---

**Algorithm 1** Meta Mirror Descent learning algorithm.

---

1: **Input:** $S_1, ..., S_n, M, k$          {Meta-train set, initial divergence parameters, minibatch size}
2: **Output:** $M$          {Learned divergence parameters}
3: **while** not converged or reached max steps **do**
4:      H= 0          {Set hypergradient accumulator}
5:      **for all** $S_i \in$ minibatch$(S_1, ..., S_n, k)$ **do**
6:          Init $\theta_i$          {Set random weights for base model}
7:          $\theta_i^* = \arg\min_\theta B_{\phi_M}(\theta, \theta_1)$ s.t. $\theta \in \arg\min_{\theta'} \mathcal{L}_S(\theta')$    {Train the base model with mirror descent}
8:          $H = H + \frac{1}{k}$hypergradient$(B_{\phi_M}, \mathcal{E}, \phi_M, \theta_i^*)$    {Obtain in Eq 13, detailed in Appx.A.12}
9:      **end for**
10:      $M = $ update$(M, H)$          {Update $M$ using SGD (or similar) update rule}
11: **end while**

---

divergence $B_{\phi_M}$, this leads to a tri-level optimisation problem with a new layer corresponding to the problem given in Eq. 1 required to complete a single mirror descent step,

$$\min_M \mathcal{E}(M, \theta_*(M)) \tag{4}$$

$$\text{s.t. } \theta_*(M) = \arg\min_\theta \mathcal{L}_S(\theta) = (\pi_{\phi_M} \circ \pi_{\phi_M} \circ ... \circ \pi_{\phi_M})(\theta_1) \tag{5}$$

$$\text{s.t. } \pi_{\phi_M}(\theta_t) = \arg\min_\theta \langle \nabla_\theta \mathcal{L}_S(\theta_t), \theta \rangle + \frac{1}{2\eta} B_{\phi_M}(\theta || \theta_t). \tag{6}$$

In the outer problem (Eq. 4), the algorithm aims to learn a divergence by optimising the meta-objective $\mathcal{E}$, which evaluates the optimiser performance. To achieve this requires getting the best response from the mid-level problem (Eq. 5) where the base model is trained from the initialisation $\theta_1$ to $\theta_*$ by recursively applying the $\pi_{\phi_M}$ operation, which is defined in the innermost problem (Eq. 15). The inner problem corresponds to performing a stochastic mirror descent update using a Bregman divergence based on $\phi_M$. Compared with the standard bilevel problems in meta-learning, introducing this third layer adds significant cost to both meta-train and meta-test stages. However, with a suitable choice of divergence, we can obtain a closed-form solution for the inner problem, thus reducing the cost to be similar to that of a standard bilevel optimisation problem. This section explains the main components of our method, which is summarised in Algorithm 1.

## 4.2 DIVERGENCE PARAMETERISATION

The paramaterisation of the divergence is important for a practical instantiation of our framework. Ideally it should be expressive enough to represent interesting optimisation dynamics, while being simple enough to provide an efficient or closed form solution to the innermost convex mirror descent optimisation. We describe a reasonable compromise as follows.

We define our Bregman divergences via the squared norm, $\phi_M(\theta) = \frac{1}{2}\theta^T M\theta$, where $M$ is a learnable symmetric positive definite matrix. In this case, the inner problem has a closed form solution,

$$\theta_{t+1} = \theta_t - \eta M^{-1} \nabla_\theta \mathcal{L}_S(\theta_t).$$

The derivation of this closed-form mapping is given in Appx. A.2. This parameterisation can be interpreted as a preconditioner, which has high representation potentialPark & Oliva (2019); Flennerhag et al. (2020), but it can be inefficient to compute $M^{-1}$ when the dimensionality of $\theta$ is high—as is typically the case for neural networks. To provide a better trade-off between capacity and efficiency, we reduce the capacity of $\phi_M$ by using block diagonal matrix $M = diag(M_1, M_2, ..., M_Z)$ with each block operating on the gradient of one layer in a neural network. We additionally represent each $M_\zeta$ by a Kronecker factorisation (Zehfuss, 1858; Hensel, 1889; Martens & Grosse, 2015),

$$M_\zeta = A_\zeta \otimes B_\zeta,$$

where $A_\zeta \in R^{\kappa \times \kappa}$ and $B_\zeta \in R^{\iota \times \iota}$ are two symmetric positive definite matrices parameterised by the column vectors $a_\zeta$ and $b_\zeta$,

$$A_\zeta = a_\zeta a_\zeta^T + I_\kappa, \qquad B_\zeta = b_\zeta b_\zeta^T + I_\iota.$$

To avoid explicitly constructing $M_i$, we make use of the fact that

$$M_\zeta^{-1} \nabla_{\theta_\zeta} \mathcal{L}_S(\theta_t) = (A_\zeta \otimes B_\zeta)^{-1} \nabla_{\theta_\zeta} \mathcal{L}_S(\theta) = B_\zeta^{-1} \nabla_{\theta_\zeta} \mathcal{L}_S(\theta)(A_\zeta^{-1})^T$$

where $\theta_\zeta$ denotes the weights in layer $\zeta$. $A_\zeta^{-1}$ and $B_\zeta^{-1}$ can be easily computed by composing the eigenvector matrices and inverting the diagonal eigenvalue matrix generated by Singular Value Decomposition from $A_\zeta$ and $B_\zeta$ respectively.

## 4.3 META-OBJECTIVE

The next step is to define the meta-objective in terms of the Bregman divergence parameters, $M$. An advantage of the SMD framework is that we have a formal notion of convergence from an initialisation $\theta_1$ to a solution $\theta_*$ (from Theorem 3.1). By defining the meta objective $\mathcal{E}(M, \theta)$ as a bound on the convergence rate, meta-learning $\phi_M$ leads to a faster optimiser. In particular, we design a meta-objective based on the quantities that appear in Theorem 3.1. Expanding the definition of the condition on the step size in the bound yields

$$\mathbb{E}_\xi[\mathcal{L}_S(\bar{\theta}; \xi) - \mathcal{L}_S(\theta_*; \xi)] \leq \frac{2B_{\phi_M}(\theta_* || \theta_1)L}{\lambda T} + 2\sigma^2. \tag{7}$$

Minimising the right-hand side of this inequality with respect to $M$ ensures that the learned optimiser finds a solution closer to a minima after $T$ iterations of training. As our goal is to optimise the expected speed of convergence on future tasks, we design a meta-objective that considers the average convergence rate over $n$ different meta-train tasks. We also discard terms that do not depend on $M$, and separate $\lambda$ into a regularisation term that allows us to control the importance of optimising the strong convexity coefficient. Thus, we define the meta-objective as

$$\mathcal{E}(M, \theta_T^{(1)}, \theta_T^{(2)}, ...) = \frac{1}{n} \sum_{i=1}^{n} B_{\phi_M}(\theta_T^{(i)} || \theta_1^{(i)}) + \frac{k}{\lambda} + g(M), \tag{8}$$

where $\theta_1^{(i)}$ and $\theta_T^{(i)}$ are the initial and final weights for the task $i$, and we leave $k$ as a hyperparameter that can be tuned heuristically . We will shortly define a regularisation function, $g(\cdot)$, which is used to prevent meta-overfitting. The strong convexity parameter is given by $\lambda = \min_\zeta v_{min}(M_\zeta)$, where $v_{min}$ denotes the smallest eigenvalue and the details are given in Appx. A.5.

We solve the outer loop optimisation problem (Eq. 4) by gradient descent using $\partial \mathcal{E}/\partial M$. This gradient computation relies on unrolling training trajectories in the inner loop where it is expensive to compute the gradient with standard reverse-mode differentiation. In this work, we apply implicit differentiation (Lorraine et al., 2020) to solve this problem, which will be discussed in Section 4.4.

**Generalisation of Convergence Rate**    Selecting the Bregman divergence parameters using a set of training tasks will result in an optimistic view of how well one would expect the learned optimiser to converge on novel tasks. That is, when Eq. 8 is evaluated on the meta-train tasks, it will provide an overly optimistic estimate of the convergence rate one can expect on future tasks. Given the relatively simple family of Bregman divergences we employ, it is possible to construct a high-confidence bound on how biased this estimate will be, and therefore provide convergence guarantees for novel tasks in terms of the meta-train objective:

**Theorem 4.1.** *Under the same conditions as Theorem 3.1, for any $M$, $\theta_1$, and $\theta_*$ such that $\|M - I\|_F \leq C$ and $\|\theta_1 - \theta_*\|_2 \leq r$, the following holds with probability at least $1 - \delta$,*

$$\mathbb{E}_S \mathbb{E}_\xi[\mathcal{L}_S(\bar{\theta}; \xi) - \mathcal{L}_S(\theta_*; \xi)] \leq \frac{1}{n} \sum_{i=1}^{n} \frac{2B_{\phi_M}(\theta_*^{(i)} || \theta_1^{(i)})}{\eta T} + 2\sigma^2 + \frac{Cr^2}{2\sqrt{n}} + 3Cr^2 \sqrt{\frac{ln(2/\delta)}{8n}}. \tag{9}$$

This tells us that the expected convergence rate on novel tasks depends on the learned divergence on training tasks, plus a complexity term depending on the Frobenius distance of the meta-learned optimiser weights $M$ from the identity. Note that restricting the diameter $r$ of the parameter space or neural networks is usually required to obtain generalisation guarantees (Bartlett et al., 2017; Long & Sedghi, 2020; Gouk et al., 2021), so this is not an unusual or counterproductive requirement. We therefore use $g(M) = \gamma \|M - I\|_F$ as a regulariser. The details for computing $\|M - I\|_F$ are given in Appendix A.4. Note that as $M$ approaches $I$, mirror descent will behave similarly to SGD.

## 4.4 META-GRADIENT COMPUTATION WITH IMPLICIT GRADIENT

We optimise the outer objective using a gradient-based optimisation method. Typical meta-learning pipelines will either unroll the sequence of updates performed by an iterative optimiser to solve the

inner problem (Finn et al., 2017), or make use of the Implicit Function Theorem (IFT) to compute derivatives w.r.t. parameters that appear in the inner objective (Lorraine et al., 2020). The number of inner loop iterations required to arrive at a solution in our use-case (i.e., many shot learning) makes the memory and computation requirements of unrolling this loop infeasible, so we use our prospective MD optimiser to find a solution and then apply the IFT method. However, the Bregman divergence parameters do not appear in the inner problem objective, which blocks the hypergradient computation through IFT. To this end, when computing the hypergradients, we use a rephrased inner objective expressed in terms of the Bregman divergence parameters. This is possible by using the property that mirror descent finds the fixed point closest to the initial guess of the network parameters, as measured by the corresponding Bregman divergence (Azizan et al., 2021). The alternative objective function is

$$\arg\min_{\theta} B_{\phi_M}(\theta, \theta_1) \tag{10}$$

$$\text{s.t. } \theta \in \arg\min_{\theta'} \mathcal{L}_S(\theta'). \tag{11}$$

Framing the constrained optimisation problem as its Lagrangian and replacing the constraints with $(\mathcal{L}_S(\theta_*) - \mathcal{L}_S(\theta))^2 = 0$, we obtain

$$\arg\min_{\theta} \sup_{\mu} B_{\phi_M}(\theta, \theta_1) + \mu(\mathcal{L}_S(\theta_*) - \mathcal{L}_S(\theta))^2, \tag{12}$$

where $\theta_*$ is the solution obtained via solving the inner problem with our prospective MD optimiser, and we note that $\theta = \theta_*$ provides us with an immediate solution to this transformed problem. This means the second term always evaluates to zero when evaluated at the solution, and we therefore only need to apply the IFT to the first term, yielding

$$\frac{\partial \mathcal{E}}{\partial M} = \frac{\partial \mathcal{E}}{\partial \theta} \left( \frac{\partial^2 B_{\phi_M}}{\partial \theta \, \partial \theta} \right)^{-1} \frac{\partial^2 B_{\phi_M}}{\partial \theta \, \partial M} \bigg|_{\phi_M, \theta_*(M)}, \tag{13}$$

where the inverse Hessian matrix can be approximated by Neumann series (Lorraine et al., 2020).

## 5 EXPERIMENTS

We evaluate our learned optimiser and compare its performance on a variety of tasks against commonly used standard optimisers including SGD, SGD-M (with momentum), Adam (Kingma & Ba, 2015) RMSProp (Tieleman & Hinton, 2012) and KFAC (Martens & Grosse, 2015), and meta-learning based methods, L2O (Andrychowicz et al., 2016), Meta-SGD (Li et al., 2017), Meta-Curvature (MetaCur) (Park & Oliva, 2019), ARUBA (Khodak et al., 2019), ARUBA++ (Khodak et al., 2019), PowerSign (Bello et al., 2017) and AddSign (Bello et al., 2017). We first explore synthetic tasks, followed by shallow neural networks on digit datasets, and then training ResNet on CIFAR-10, before finally evaluating deep neural networks on high resolution images.

**Algorithm deployment pipeline:** For each set of experiments, we train MetaMD on a set of meta-train datasets, and evaluate it on a disjoint set of meta-test datasets. In the meta-test stage, models are trained by MetaMD (or competitors) using each dataset's standard training set, and evaluated on the corresponding test splits. We emphasize that for meta-testing, each optimiser consumes the same amount of data, and a comparable amount of computing per iteration, except KFAC is much more expensive due to its inverse operation in every iteration. The learned optimisers such as MetaMD, MetaSGD, MetaCur, and X-Sign use additional data and compute for the prior meta-training stage, but this is a one-off cost that can be amortized across different meta-test problems of interest.

### 5.1 SYNTHETIC PROBLEM: META-QUADRATIC OPTIMISATION

We start with a synthetic experiment to illustrate the impact of the proposed meta objective, and simply using exact Forward Mode Differentiation (FMD) for hypergradient for learning. Meta Mirror Descent is deployed on a family of 2D quadratic optimisation problems from which we can sample a disjoint set of meta-training and meta-testing optimisation problems. We sample tasks of the form:

$$\min_{\theta} \theta^T Q \theta - b^T \theta$$

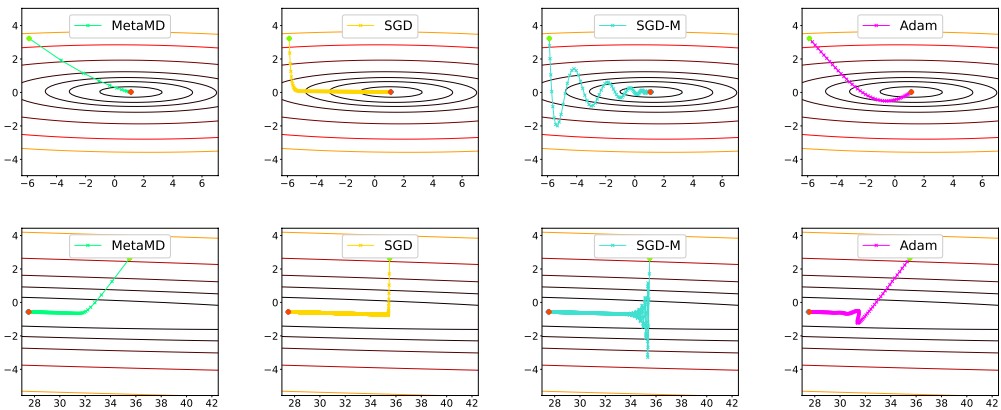

Figure 1: Trajectory comparison of different optimisers on two quadratic optimisation problems (rows). Columns represent optimisers from left to right: MetaMD, SGD, SGD-M and Adam. The point green point denotes the starting point and the orange point denotes the minima. Top row: Iterations to convergence are 23, 928, 104 and 45 respectively. Bottom Row: Iterations to convergence are 280, 27,328, 1,838 and 324 respectively. MetaMD is the fastest.

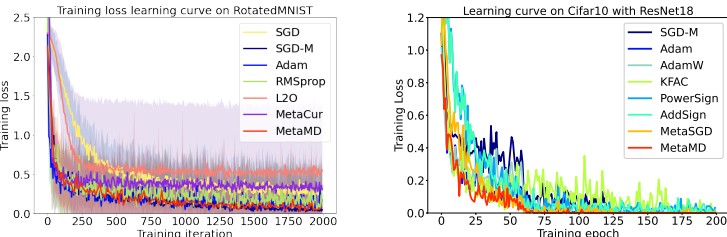

Figure 2: Comparison of training loss curves for different optimisers. Left: RotatedMNIST, averaged over all held-out domains. Right: Training loss curve of ResNet18 trained on CIFAR10

where $Q$ and $b$ are random variables. We give the details of the setting in Appx. A.1. A comparison of optimisation trajectories on two kinds of meta-test quadratic problems is shown in Fig. 1. All the optimisers are initialised in the same position and stopped when the stop criterion is met. All the optimisers reach the minima, but those that have not meta-learned the curvature information in this family of problems require more iterations to converge compared with MetaMD.

## 5.2 LEARNING MIRROR DESCENT FOR NEURAL NETWORKS

**RotatedMNIST and MLPs** We first evaluate optimiser learning for neural networks using the RotatedMNIST dataset and a 3-layer MLP architecture. RotatedMNIST defines 6 domains by rotating the original MNIST dataset by 0, 15, 30, 45, 60 and 75 degrees. We use 5 domains for meta-training, and train MetaMD to convergence in the inner loop, and evaluate the performance on the held-out domain. This process is repeated, holding out each domain in turn as meta-test. The convergence curve is shown in Fig. 2(left), and the testing performance in Table 1(top). We can see that MetaMD converges rapidly and trains models with strong testing performance. The hyperparameter tuning protocol for this and other experiments in this section is explained in Appx A.7.

**Diverse Digit Datasets and Small CNNs** Next we explore applying MetaMD to a more diverse set of datasets and CNN classifiers. We collect a set of datasets, which we denote as DiverseDigits. They include: MNIST LeCun & Cortes (2010), QMNIST Yadav & Bottou (2019), KMNIST Clanuwat et al. (2018), FashionMNIST Xiao et al. (2017), USPS Hull (1994) and SVHN Netzer et al. (2011). We train LeNet classifier using MetaMD, resizing all images to $28 \times 28$ greyscale. The same leave-one-dataset-out protocol is used: Each dataset is held out in turn for evaluation after MetaMD is trained on the other datasets. Compared to the previous RotatedMNIST experiment, the distribution

Table 1: Test Accuracy (%) on RotatedMNIST and DiverseDigits with 3-layer MLP and LeNet respectivly. Each column is a test dataset, and MetaMD is trained on the other datasets.

| | Test domain | 0 | 15 | 30 | 45 | 60 | 75 |
|---|---|---|---|---|---|---|---|
| **3-Layer MLP** | SGD | $92.23 \pm 0.57$ | $91.91 \pm 0.49$ | $92.57 \pm 0.32$ | $92.89 \pm 0.35$ | $92.73 \pm 0.32$ | $92.36 \pm 0.87$ |
| | SGD-M | $94.77 \pm 0.58$ | $94.64 \pm 0.14$ | $94.66 \pm 0.29$ | $94.67 \pm 0.47$ | $94.60 \pm 0.47$ | $94.47 \pm 0.63$ |
| | Adam | $92.96 \pm 0.58$ | $93.29 \pm 0.92$ | $93.51 \pm 0.84$ | $93.69 \pm 0.99$ | $93.67 \pm 0.35$ | $92.98 \pm 1.17$ |
| | RMSprop | $92.48 \pm 0.49$ | $93.56 \pm 0.51$ | $92.77 \pm 0.50$ | $93.58 \pm 0.32$ | $93.43 \pm 0.32$ | $93.14 \pm 0.31$ |
| | L2O | $93.48 \pm 0.74$ | $93.46 \pm 0.44$ | $93.44 \pm 0.61$ | $93.84 \pm 0.51$ | $93.04 \pm 0.22$ | $93.64 \pm 0.23$ |
| | MetaCur | $93.92 \pm 0.45$ | $94.62 \pm 0.71$ | $94.98 \pm 0.30$ | $94.43 \pm 0.43$ | $94.98 \pm 0.71$ | $93.83 \pm 0.92$ |
| | MetaMD | $\mathbf{95.67} \pm 0.39$ | $\mathbf{95.59} \pm 0.28$ | $\mathbf{95.50} \pm 0.61$ | $\mathbf{95.33} \pm 0.45$ | $\mathbf{95.03} \pm 0.63$ | $\mathbf{94.99} \pm 0.52$ |
| | Test domain | MNIST | QMNIST | KMNIST | FashionMNIST | USPS | SVHN |
| **LeNet** | SGD | $96.44 \pm 0.91$ | $96.23 \pm 0.73$ | $87.61 \pm 1.87$ | $\mathbf{88.95} \pm 1.43$ | $92.73 \pm 1.13$ | $85.44 \pm 1.22$ |
| | SGD+M | $98.47 \pm 0.16$ | $97.21 \pm 0.15$ | $92.54 \pm 0.62$ | $86.44 \pm 0.45$ | $95.37 \pm 0.24$ | $86.26 \pm 0.48$ |
| | Adam | $98.49 \pm 0.17$ | $98.10 \pm 0.33$ | $93.20 \pm 0.82$ | $87.36 \pm 0.55$ | $93.68 \pm 0.38$ | $\mathbf{87.07} \pm 0.61$ |
| | RMSprop | $98.65 \pm 0.21$ | $98.30 \pm 0.09$ | $93.14 \pm 0.87$ | $87.45 \pm 0.13$ | $95.43 \pm 1.06$ | $87.01 \pm 0.18$ |
| | L2O | $90.80 \pm 0.27$ | $92.60 \pm 0.34$ | $87.27 \pm 0.68$ | $82.34 \pm 0.66$ | $91.23 \pm 0.54$ | $34.42 \pm 0.43$ |
| | MetaCur | $97.45 \pm 0.48$ | $97.62 \pm 0.67$ | $93.74 \pm 0.27$ | $87.50 \pm 0.58$ | $93.42 \pm 0.67$ | $86.27 \pm 0.53$ |
| | MetaMD | $\mathbf{98.72} \pm 0.33$ | $\mathbf{98.74} \pm 0.24$ | $\mathbf{96.56} \pm 0.49$ | $87.33 \pm 0.36$ | $\mathbf{95.96} \pm 0.14$ | $86.43 \pm 0.64$ |

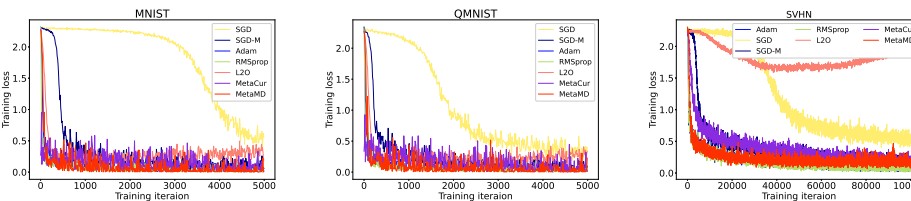

Figure 3: Learning curves for DiverseDigits. From left to right: MNIST, QMNIST and SVHN.

of tasks used for meta-training and meta-testing is now more diverse and challenging. Due to the greater cost of training the base model here, we use $T = 500$ iterations for the inner loop, and leave efficient meta-learning under longer-horizons as future work. We compare all methods fairly using a common hyper-parameter tuning protocol for meta-test. Specifically, we perform grid search with respect to meta-test validation accuracy for each competitor, and more detail is given in Appx. A.7.

The results averaged over 3 meta-test trials are shown as testing performance at convergence in Table 1(bottom) and selected meta-test learning curves in Fig. 3, with the remaining learning curves given in Appx. A.8. We can see that MetaMD is clearly faster than SGD and SGD-M in training convergence (Fig. 3), while typically producing models with the strongest generalisation error (Table 1). It is noteworthy that MetaMD exhibits strong cross-dataset generalisation here, corroborating our Theorem 4.1 on cross-task optimiser generalisation.

**Application to ResNet18 and CIFAR10**   We next explore scaling up to deeper architectures, aiming to train a ResNet18 on CIFAR10 as a held out testing task. To this end, we construct a suite of meta-training datasets by combining STL10 (Coates et al., 2011) and DiverseDigits from the previous setting. ResNet18+CIFAR10 is a well-studied problem with lots of known tuning tricks for standard optimisers. For fair comparison, we therefore tune all methods with exactly the same BayesOpt-based hyperparameter tuning protocol, based on CIFAR10 validation performance. For this experiment we also compare some other learned optimisers, including: (i) PowerSign and AddSign that are not directly comparable. They were trained on CIFAR10 (for us it is a held out dataset), but on a different architecture Bello et al. (2017). (ii) KFAC which is directly comparable insofar as using the same structure of preconditioner to MetaMD, but not comparable in that it computes the preconditioner during meta-test via the Fisher Information Matrix of the target problem – while MetaMD amortizes this cost and meta-learns it from a disjoint set of tasks, without tuning to the target task. (iv) MetaSGD which we re-implement to use exactly the same meta-train data as MetaMD for direct comparability.

Table 2: Test accuracy on CIFAR10 using Resnet18 and various optimisers

| Method | KFAC | SGD-M | Adam | AdamW | PowerSign | AddSign | MetaSGD | MetaMD |
|---|---|---|---|---|---|---|---|---|
| Accuracy | $87.54 \pm 0.23$ | $91.37 \pm 0.45$ | $91.64 \pm 0.56$ | $92.66 \pm 0.32$ | $87.77 \pm 0.32$ | $88.29 \pm 0.67$ | $91.54 \pm 0.17$ | $\mathbf{93.56} \pm 0.34$ |

Table 3: Test Accuracy (%) on high resolution datasets. Comparison with various optimiser.

| Datasets | Caltech | DTD | Flowers | Pubfig |
|----------|---------|-----|---------|--------|
| SGD-M | $26.95 \pm 0.35$ | $32.66 \pm 0.61$ | $47.18 \pm 0.17$ | $74.10 \pm 0.11$ |
| Adam | $27.21 \pm 0.53$ | $28.09 \pm 0.44$ | $41.03 \pm 0.48$ | $73.31 \pm 0.18$ |
| PowerSign | $26.11 \pm 0.39$ | $31.91 \pm 0.31$ | $44.66 \pm 0.53$ | $68.25 \pm 0.42$ |
| AddSign | $27.21 \pm 0.37$ | $29.52 \pm 0.24$ | $44.79 \pm 0.47$ | $67.77 \pm 0.29$ |
| Meta-SGD | $25.41 \pm 0.58$ | $27.93 \pm 0.34$ | $48.77 \pm 0.44$ | $66.51 \pm 0.36$ |
| MetaCur | $12.19 \pm 2.04$ | $15.96 \pm 1.42$ | $6.46 \pm 0.65$ | $28.98 \pm 1.87$ |
| ARUBA | $21.84 \pm 0.74$ | $21.17 \pm 1.31$ | $20.26 \pm 0.79$ | $52.35 \pm 1.16$ |
| ARUBA++ | $24.26 \pm 0.97$ | $25.64 \pm 0.64$ | $39.24 \pm 0.67$ | $67.29 \pm 0.63$ |
| MetaMD | $\mathbf{37.50} \pm 1.59$ | $\mathbf{41.01} \pm 0.87$ | $\mathbf{54.84} \pm 1.62$ | $\mathbf{77.77} \pm 0.87$ |

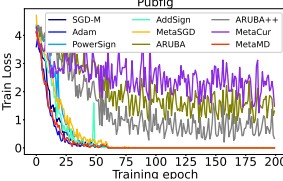 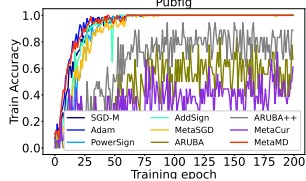 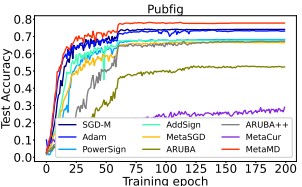

Figure 4: Learning curves on Pubfig. The left, middle and right column represent Training loss, Training accuracy and Test accuracy respectively.

From the results in Fig. 2 (right), we can see that MetaMD converges rapidly as expected; and from the results in Table 2 we can see that it also provides the best generalisation performance.

**High Resolution Image Application**     Finally, we explore benchmarks with higher resolution images with ResNet18. We construct a set of datasets including Aircraft (Maji et al., 2013), Butterfly (Chen et al., 2018), Pets (Parkhi et al., 2012), Caltech (Griffin et al., 2007), DTD (Cimpoi et al., 2014), Flowers (Nilsback & Zisserman, 2008), Pubfig (Pinto et al., 2011). See Appx. A.9 for the details about each. Compared with the ($28\times$ and $32\times$) images used in the previous experiment settings, these datasets are much higher resolution images. We resize them to standard $224 \times 224$ for ResNet18. We divide this suite of datasets into meta-train and meta-test groups. Meta-train contains Aircraft, Butterfly and Pets. Meta-test has Caltech, DTD, Flowers, and Pubfig. Compared with Rotated MNIST and Diverse Digit settings, the meta-train and meta-test domains are much more different, which affects the competitors whose initialisations are part of the meta-training objectives. MetaMD and the competitors including ARUBA, ARUBA++, MetaCur and Meta-SGD, are trained on the meta-train domains. MetaMD requires the model trained to be converged in each outer loop iteration. To improve meta-training efficiency, therefore we train the base model on 20 subcategories of each domain which are randomly sampled in each outer loop iteration. All the competitors are tested on the meta-test tasks by training ResNet18 for each dataset individually. We did fair hyperparameter tuning for every competitor as detailed in Appx. A.7. We compare the learning curve and the test set performance of the learned model on each dataset. The learning curves on Pubfig are shown in Fig. 4 and the others are in Appx. A.9. Online algorithms ARUBA and ARUBA++ converge poorly in terms of the training loss during meta-test. This is due to their prior poor convergence during meta-train, in contrast to the batch competitors that better exploit the meta-train tasks. In contrast, MetaCur converged well during meta-train, but suffers from meta-overfitting and converges poorly during meta-test. Meanwhile, MetaMD exhibits rapid convergence as expected. In terms of test accuracy on the downstream tasks, Table 3 shows that MetaMD performs best, thanks to knowledge transfer from the source tasks reducing overfitting to these comparatively small target tasks.

# 6 CONCLUSION

We explored meta-learning neural network optimisers from the Mirror Descent perspective – in practice by meta-learning a Bregman Divergence to manipulate the gradient during learning. Our approach has clear theoretical motivation by optimizing a bound on the convergence rate, and has both a convergence guarantee and a cross-dataset generalisation guarantee. Our practical implicit-gradient based algorithm surprisingly (Lorraine et al., 2020) enables meta-learning of learning rates via implicit gradient, and is empirically fast and effective compared to SGD and other competitors.

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

## A APPENDIX

### A.1 QUADRATIC SETTING

Meta Mirror Descent is deployed on a family of 2D quadratic optimisation problems from which we can sample a disjoint set of meta-training and meta-testing optimisation problems. We sample tasks of the form:

$$\min_\theta \theta^T Q\theta - b^T \theta$$

where $Q$ and $b$ are random variables. $b$ follows a Gaussian distribution with mean vector $[1,1]^T$ and identity covariance. To generate $Q$, which controls the flatness of the quadratic function, we sample $Q_{0,0}$ and $Q_{1,1}$, independently with different Gaussian distributions. For example, on the top row in Figure 1: the mean of $Q_{0,0}$ and $Q_{1,1}$ are 0.3 and 14 while on the bottom row, the mean of $Q_{0,0}$ and $Q_{1,1}$ are 0.02 and 14 respectively. In addition, the initialisation of each optimisation problem is also a random variable.

### A.2 DERIVE OF THE CLOSED FORM MIRROR LOOP

In our setting, the mirror loop is described as

$$\theta_{t+1} = \arg\min_\theta \langle \nabla_\theta \mathcal{L}(\theta_t), \theta \rangle + \frac{1}{2\eta} B_\phi(\theta||\theta_t) \tag{14}$$

or, equivalently,

$$\theta_{t+1} = \arg\min_\theta \eta \langle \nabla_\theta \mathcal{L}_{tr}(\theta_t), \theta \rangle + B_\phi(\theta||\theta_t). \tag{15}$$

Setting the gradient w.r.t. $\theta$ to zero, we have

$$\eta \nabla \mathcal{L}(\theta_t) + \nabla\phi(\theta_{t+1}) - \nabla\phi(\theta_t) = 0, \tag{16}$$

which when rearranged yields

$$\nabla\phi(\theta_{t+1}) = \nabla\phi(\theta_t) - \eta\nabla\mathcal{L}(\theta_t) \tag{17}$$

$$\theta_{t+1} = \nabla\phi^{-1}(\nabla\phi(\theta_t) - \eta\nabla\mathcal{L}(\theta_t)). \tag{18}$$

In our case

$$\phi(\theta) = \frac{1}{2}\theta^T M\theta, \tag{19}$$

where $M$ is a block diagonal matrix. Therefore,

$$\nabla\phi(\theta) = M\theta \tag{20}$$

$$\nabla\phi^{-1}(\theta) = M^{-1}\theta. \tag{21}$$

As a result, we also have

$$\theta_{t+1} = \nabla\phi^{-1}(\nabla\phi(\theta_t) - \eta\nabla\mathcal{L}(\theta_t)) \tag{22}$$

$$= M^{-1}(M\theta_t - \eta\nabla\mathcal{L}(\theta_t)) \tag{23}$$

$$= \theta_t - \eta M^{-1}\nabla\mathcal{L}(\theta_t). \tag{24}$$

### A.3 PROOF OF THEOREM 4.1

We will make use of a well-known bound on suprema of empirical processes due to Bartlett & Mendelson (2002).

**Theorem A.1.** *For all functions $f \in \mathcal{F}$ with $\|f\|_\infty \leq a$, and i.i.d. $Z_i$, the following holds with probability at least $1 - \delta$,*

$$\mathbb{E}[f(Z)] \leq \frac{1}{n}\sum_{i=1}^n f(Z_i) + 2\hat{R}_n(\mathcal{F}) + 3a\sqrt{\frac{ln(2/\delta)}{2n}}. \tag{25}$$

In this theorem, $\hat{R}_n(\mathcal{F})$ is the empirical Rademacher complexity of the class $\mathcal{F}$, defined as

$$\hat{R}_n(\mathcal{F}) = \mathbb{E}_\epsilon \left[ \sup_{f \in \mathcal{F}} \frac{1}{n} \sum_{i=1}^{n} \epsilon_i f(Z_i) \right], \tag{26}$$

where $\epsilon_i$ is a Rademacher random variable, so $P(\epsilon_i = -1) = P(\epsilon_i = 1) = 0.5$.

*Proof.* It suffices to bound, with high confidence, the difference between the first term of the meta-objective, and the expected Bregman divergence between initializations and solutions on new tasks sampled from the same task distribution. We will obtain such a bound using Rademacher complexity, and the main result will follow from standard applications of Rademacher complexity-based generalisation bounds (Bartlett & Mendelson, 2002), along with the observation that $B_\phi(\theta_* \| \theta_1) \leq \frac{Cr^2}{2}$. That is, we will treat the initial and final points of an optimisation trajectory for each task as a random variable, and we will bound to what extent the mean Bregman divergence between initial and trained parameters on some meta-train tasks can deviate from the expected Bregman divergence on unseen tasks. In particular, we analyse the following class:

$$\mathcal{F} = \{(\theta_*, \theta_1) \mapsto B_\phi(\theta_* \| \theta_1) \; : \; \phi(\theta) = \frac{1}{2}\theta^T M \theta, \; \|M - I\|_F \leq C\}. \tag{27}$$

Denoting $\theta_*^{(i)} - \theta_1^{(i)}$ by $\tilde{\theta}_i$, we can bound the Rademacher complexity of this class from above by

$$\hat{R}_n(\mathcal{F}) = \mathbb{E}_\epsilon \left[ \sup_{B_\phi \in \mathcal{F}} \frac{1}{n} \sum_{i=1}^{n} \epsilon_i B_\phi(\theta_*^{(i)} \| \theta_1^{(i)}) \right] \tag{28}$$

$$= \frac{1}{2n} \mathbb{E}_\epsilon \left[ \sup_{M} \sum_{i=1}^{n} \epsilon_i \tilde{\theta}_i^T M \tilde{\theta}_i \right] \tag{29}$$

$$= \frac{1}{2n} \mathbb{E}_\epsilon \left[ \sup_{M} \sum_{i=1}^{n} \epsilon_i \langle M, \tilde{\theta}_i \tilde{\theta}_i^T \rangle_F \right] \tag{30}$$

$$= \frac{1}{2n} \mathbb{E}_\epsilon \left[ \sup_{M} \langle M, \sum_{i=1}^{n} \epsilon_i \tilde{\theta}_i \tilde{\theta}_i^T \rangle_F \right] \tag{31}$$

$$= \frac{1}{2n} \mathbb{E}_\epsilon \left[ \sup_{M} \langle (M - I), \sum_{i=1}^{n} \epsilon_i \tilde{\theta}_i \tilde{\theta}_i^T \rangle_F \right] + \frac{1}{2n} \mathbb{E}_\epsilon \left[ \langle I, \sum_{i=1}^{n} \epsilon_i \tilde{\theta}_i \tilde{\theta}_i^T \rangle_F \right], \tag{32}$$

where $\langle \cdot, \cdot \rangle_F$ is the Frobenius inner product. Note that the second term in the final equality is equal to zero. As such we can continue by further bounding the first term using the Cauchy-Schwarz inequality,

$$\hat{R}_n(\mathcal{F}) \leq \frac{C}{2n} \mathbb{E}_\epsilon \left[ \left\| \sum_{i=1}^{n} \epsilon_i \tilde{\theta}_i \tilde{\theta}_i^T \right\|_F \right] \tag{33}$$

The remainder of the proof follows a well-known sequence of steps using when bounding the expected norm of a Rademacher sum (see, e.g., (Shalev-Shwartz & Ben-David, 2014)), which we include here for completeness. Jensen's inequality tells us that

$$\mathbb{E}_\epsilon \left[ \left\| \sum_{i=1}^{n} \epsilon_i \tilde{\theta}_i \tilde{\theta}_i^T \right\|_F \right] \leq \sqrt{\mathbb{E}_\epsilon \left[ \left\| \sum_{i=1}^{n} \epsilon_i \tilde{\theta}_i \tilde{\theta}_i^T \right\|_F^2 \right]} \tag{34}$$

$$= \sqrt{\mathbb{E}_\epsilon \left[ \sum_{i=1}^{n} \sum_{j=1}^{n} \epsilon_i \epsilon_j \langle \tilde{\theta}_i \tilde{\theta}_i^T, \tilde{\theta}_j \tilde{\theta}_j^T \rangle_F \right]}. \tag{35}$$

Noting that $\mathbb{E}[\epsilon_i \epsilon_j]$ is one if $i = j$ and zero otherwise, we obtain

$$\sqrt{\mathbb{E}_\sigma\left[\sum_{i=1}^n \sum_{j=1}^n \epsilon_i \epsilon_j \langle \tilde{\theta}_i \tilde{\theta}_i^T, \tilde{\theta}_j \tilde{\theta}_j^T \rangle_F\right]} \leq \sqrt{\sum_{i=1}^n \|\tilde{\theta}_i \tilde{\theta}_i^T\|_F^2} \tag{36}$$

$$\leq \sqrt{\sum_{i=1}^n (\|\tilde{\theta}_i\|_2 \|\tilde{\theta}_i\|_2)^2} \tag{37}$$

$$\leq \sqrt{nr^4}. \tag{38}$$

Substituting this back into our earlier derivation yields

$$\hat{R}_n(\mathcal{F}) \leq \frac{C\sqrt{nr^4}}{2n} \tag{39}$$

$$= \frac{Cr^2}{2\sqrt{n}}, \tag{40}$$

which concludes the proof. $\qquad\square$

## A.4 META REGULARISER COMPUTATION

We provide the meta regualriser computation details in this section. The proposed meta regulariser is expressed as

$$\|M - I\|_F^2 = \|M\|_F^2 + \| - I\|_F^2 + 2\langle M, -I \rangle_F \tag{41}$$

$$= \|M\|_F^2 + \|I\|_F^2 - 2\text{tr}(M) \tag{42}$$

$$= \|M\|_F^2 + \|I\|_F^2 - 2\text{tr}(A)\text{tr}(B) \tag{43}$$

$$= \sum_{i,j} \upsilon_i(A)\upsilon_j(B) + \|I\|_F^2 - 2\text{tr}(A)\text{tr}(B), \tag{44}$$

where $\upsilon_i(\cdot)$ denotes the $i$-th singular value of a matrix. To get from Eq. 42 to Eq. 43 we use that $\text{tr}(A \otimes B) = \text{tr}(A)\text{tr}(B)$, and from Eq. 43 to Eq. 44 we use that $\|M\|_F^2 = \sum_i \upsilon_i^2(A \otimes B) = \sum_{i,j} \upsilon_i(A)\upsilon_j(B)$.

## A.5 COMPUTING $\lambda$

In this section, we provide the computation details for $\lambda$, as used in Equation 8:

$$\lambda = \min_\zeta \upsilon_{min}(M_\zeta) \tag{45}$$

$$= \min_\zeta \upsilon_{min}(A_\zeta \otimes B_\zeta) \tag{46}$$

$$= \min_\zeta \min_{i,j}(\upsilon_i(A_\zeta)\upsilon_j(B_\zeta)) \tag{47}$$

where $\upsilon_i(\cdot)$ denotes the $i$-th singular value of a matrix. From Eq. 46 to Eq. 47, we use that $\upsilon(A \otimes B) = \upsilon(A) \cdot \upsilon(B)^T$, where $\upsilon(\cdot)$ gives all the eigenvalues in the column vector form.

## A.6 GRADIENT COMPUTATION FOR DIAGONAL MATRIX

Diagonal matrix is a special case of block diagonal matrix, which we applied in the 2D quadratic setting in section 5.2. With this simple parameterisation, the exact hypergradient w.r.t. the target optimiser can be computed with Forward Mode Differential (FMD). The gradient of the second term in the proposed meta-objective in Eq.8 is easy to compute while the first term with respect to $\phi$ is expressed as:

$$\frac{\partial B_{\phi_M}(\theta_*, \theta_1)}{\partial M} \approx \frac{\partial B_{\phi_M}(\theta_T, \theta_1)}{\partial M} \tag{48}$$

$$= \underbrace{\frac{\partial B_{\phi_M}(\theta_T, \theta_1)}{\partial M}}_{\text{direct gradient}} + \underbrace{\frac{\partial B_{\phi_M}(\theta_T, \theta_1)}{\partial \theta_T} \frac{\partial \theta_T}{\partial M}}_{\text{indirect gradient}} \tag{49}$$

when T is large enough to satisfy that $\theta_* \approx \theta_T$. The computation of the direct gradient can be easily solved by the existing auto-differentiation library. The indirect gradient in Eq 49, usually termed hypergradient, is much more computationally chanllenging as it is expressed in the form:

$$\frac{\partial \theta_T}{\partial M} = \sum_{t=1}^{T} \left( \prod_{t'=t+1}^{T} A_{t'} \right) B_t \tag{50}$$

$$\text{s.t. } A_t = \frac{\partial \pi_{\phi_M}(\theta_{t-1})}{\partial \theta_{t-1}} = I - \eta M^{-1} \frac{\partial^2}{\partial \theta^2} \mathcal{L}_{tr}(\theta_{t-1}), \tag{51}$$

$$B_t = \frac{\partial \pi_{\phi_M}(\theta_{t-1})}{\partial M} = 2\eta \frac{\partial}{\partial \theta} \mathcal{L}_{tr}(\theta_{t-1}) M^{-2}. \tag{52}$$

Forward-Mode Differentiation (FMD) and Reverse-Mode Differentiation Franceschi et al. (2017) are two algorithms to compute Eq 50. RMD computes the gradient from the last to the initial step, requiring one to store the entire optimisation trajectory in memory. When the optimisation trajectory is long, this computation is very expensive. In comparison, FMD updates the hypergradient in parallel with in inner loop optimisation by:

$$\frac{\partial \theta_t}{\partial M} = \frac{\partial \pi_{\phi_M}(\theta_{t-1})}{\partial \theta_{t-1}} \frac{\partial \theta_{t-1}}{\partial M} + \frac{\partial \pi_{\phi_M}(\theta_{t-1})}{\partial M}, \tag{53}$$

where it only requires the information from step $t-1$.

### A.7 HYPERPARAMETER TUNING

**Grid Search**  For tuning the hyperparameters on 3-layer MLPs model settings in section 5.2, we sweep over the learning rates $\{0.1, 0.05, 0.01, 0.005, 0.001\}$ and weight decay parameters of $\{0.001, 0.0001, 0.0005\}$ for the SGD, SGD-M and RMSprop. In terms of Adam, we do grid search over the learn rates $\{0.3, 0.2, 0.1, 0.01, 0.001\}$ and weight decay $\{0.001, 0.0001, 0.0005\}$. For L2O, MetaCur, ARUBA and ARUBA++, we tune the models on the meta-test setting with the learning rates $\{0.1, 0.01, 0.005, 0.001, 0.0005, 0.0001\}$ and weight decay $\{0.001, 0.0001, 0.0005\}$.

**Bayesian Optimisation**  We implement our BayesOpt using (Balandat et al., 2020) for the hyperparameter tuning on the ResNet18 and CIFAR10 setting. The model the expected performance using a Gaussian process with RBF kernel, which maps the learning rate and weight decay to the estimated validation accuracy. This also provides uncertainty information to the Upper Confidence Bound (UCB) acquisition function for exploring/exploiting the hyperparameter space. For each model selection in the meta-test stage, we run the Bayesian optimisation for 25 iterations.

### A.8 TRAINING LOSS LEARNING CURVE FOR DIVERSEDIGITS DATASET

We give all the training loss learning curves on DiverseDigits in Fig 5. It can be noticed that the conclusion we drew that MetaMD is clearly faster than SGD and SGD-M in training convergence in Section 5.2 is further supported.

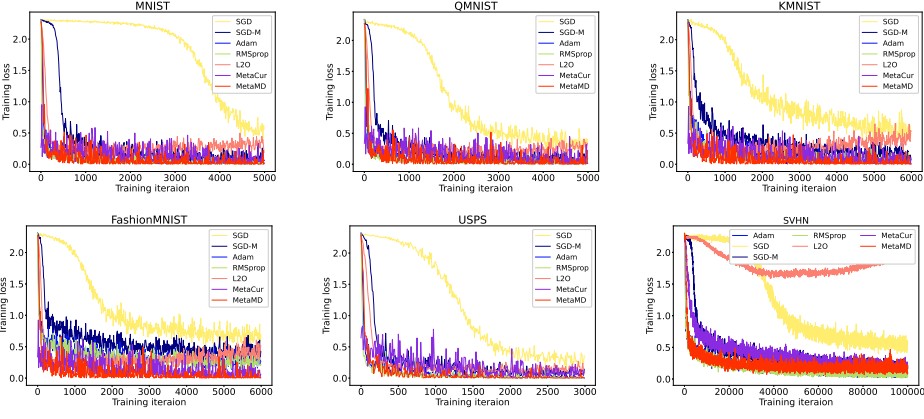

Figure 5: Convergence comparison of different optimisers on DiverseDigits.

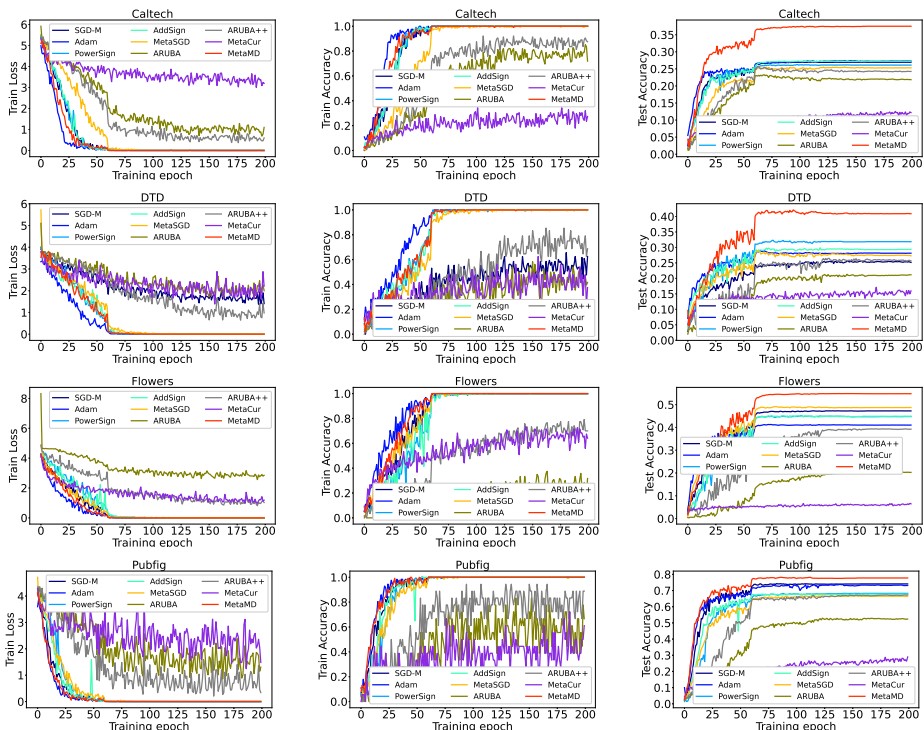

Figure 6: Learning curves on Caltech, DTD, Flowers and Pubfig. The left, middle and right collumn represent Training loss, Training accuracy and Test accuracy respectively

## A.9 EXPERIMENT DETAILS ON HIGH RESOLUTION IMAGE DATASETS

We provide a summary of each dataset applied in our High Resolution tasks and give a clear split for the meta-train and meta-test datasets in Table 4. All the datasets except Caltech-256 has standard training, testing splits. As a result, we build our own split with 60 and 20 examples for each classes in training and testing stage respectively. Both MetaMD and competitors are tuned on learning rates,

Table 4: Statistics for the datasets used throughout the experiments. The Train, and Test columns contain the number of instances in each of the corresponding subsets.

|  | Dataset | Train | Test | Classes |
|---|---|---|---|---|
| Meta-Train | Aircraft (Maji et al., 2013) | 6,667 | 3,333 | 100 |
|  | Butterfly (Chen et al., 2018) | 10,270 | 15,009 | 200 |
|  | Pets (Parkhi et al., 2012) | 4,000 | 3,390 | 37 |
| Meta-Test | Caltech (Griffin et al., 2007) | 12,800 | 5,120 | 256 |
|  | DTD (Cimpoi et al., 2014) | 3,760 | 1,880 | 47 |
|  | Flowers (Nilsback & Zisserman, 2008) | 2,040 | 6,149 | 102 |
|  | PubFig (Pinto et al., 2011) | 12,178 | 1,660 | 83 |

$\{0.1, 0.01, 0.001, 0.0001\}$ and weight decays, $\{0.001, 0.0005, 0.0001, 0.00001, 0\}$. The learning curve for Caltech, DTD, Flowers and Pubfig are shown in Fig 6.

## A.10 LOSS LANDSCAPE ANALYSIS

The generalisation ability is reflected by the flatness of the converged loss landscape (Foret et al., 2021). Motivated by this, we compare the converged loss landscapes achieved by MetaMD with those by SGD on the High Resolution Image setting. From Figure 7, it can be observed that the landscapes reached by MetaMD are much more flattened than SGD.

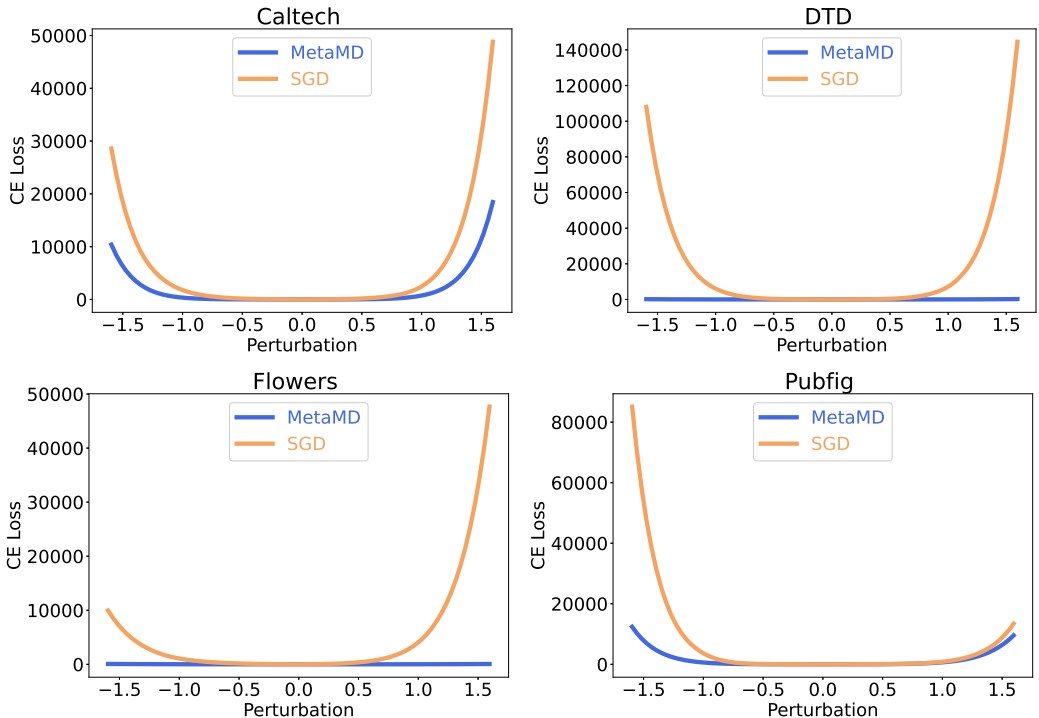

Figure 7: 1D Training Loss Landscape on Caltech, DTD, Flowers and Pubfig, where perturbations are performed in the direction of the eigenvector of the loss Hessian matrix corresponding to the largest eigenvalue.

### A.11 LOW RANK REPRESENTATION ANALYSIS

The recent research (Saunshi et al., 2021; Chen & Lee, 2021) claim that low rank representation generalizes well. Following the low-rank representation analysis proposed in (Chen & Lee, 2021), we compare the learned feature space of the model trained by SGD and that trained by MetaMD in the high resolution setting in Fig. 8. $-\sum_i \tilde{\lambda}_i \log \tilde{\lambda}_i$ denotes the low rank level, where the smaller value represents the lower rank and $\tilde{\lambda}_i = \lambda_i / \lambda_{max}$ represent the normalized singular values. One can observe that MetaMD has the lower rank feature spaces on all the meta-test tasks than SGD. The decay speed of the normalized singular values also indicates that MetaMD learns the lower rank feature space than SGD.

### A.12 HYPERGRADIENT COMPUTATION

In this section, we introduce details of the inverse Hessian computation. The hypergradient of the meta-objective $\mathcal{E}$ w.r.t. to the optimiser parameters $M$ is computed through

$$\frac{\partial \mathcal{E}}{\partial M} = \frac{\partial \mathcal{E}}{\partial \theta} \left( \frac{\partial^2 B_{\phi_M}}{\partial \theta \, \partial \theta} \right)^{-1} \frac{\partial^2 B_{\phi_M}}{\partial \theta \, \partial M} \Bigg|_{\phi_M, \theta_*(M)} = \lim_{i \to \infty} \frac{\partial \mathcal{E}}{\partial \theta} \sum_{j=0}^{i} \left( I - \frac{\partial^2 B_{\phi_M}}{\partial \theta \, \partial \theta} \right)^{j} \frac{\partial^2 B_{\phi_M}}{\partial \theta \, \partial M} \Bigg|_{\phi_M, \theta_*(M)},$$

where following Lorraine et al. (2020), the inverse Hessian matrix is approximated by Neumann series:

$$\left( \frac{\partial^2 B_{\phi_M}}{\partial \theta \, \partial \theta} \right)^{-1} = \lim_{i \to \infty} \sum_{j=0}^{i} \left( I - \frac{\partial^2 B_{\phi_M}}{\partial \theta \, \partial \theta} \right)^{j}.$$

In practice, the approximation iteration $j$ does not need to go to infinity, and in our case we set it as 20. We give the implementation details for computing $\frac{\partial \mathcal{E}}{\partial M}$ in Algorithm 2 by adapting the algorithm proposed in (Lorraine et al., 2020) to our problem, where $\alpha$ is the hyperparameter controlling the iteration step length.

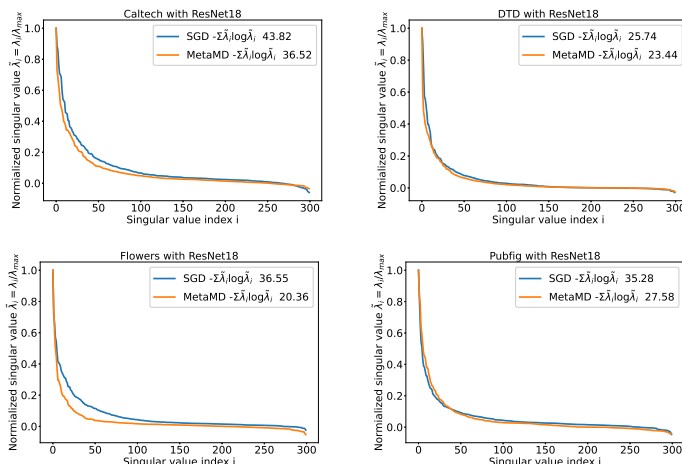

Figure 8: Low rank representation Analysis on Caltech, DTD, Flowers and Pubfig.

---

**Algorithm 2** Computing the hypergradient of the meta-objective $\mathcal{E}$, with respect to the optimiser parameter $M$. The $\text{grad}(\cdot, \cdot, \cdot)$ function from PyTorch computes a Jacobian-vector product when called with a non-scalar first argument. Inspired by Lorraine et al. (2020), we use this to efficiently compute the Hessian required for approximating the Neumann series.

---

**Input:** $B_{\phi_M}, \mathcal{E}, \phi_M, \theta_i^*$
**Output:** $-p\frac{\partial^2 B_{\phi_M}}{\partial\theta\partial M}$
$v = p = \frac{\partial\mathcal{E}}{\partial\theta}|_{\phi_M, \theta_i^*}$
**for all** $j = 1, ..., J$ **do**
  $v- = \alpha \cdot \text{grad}(\frac{\partial B_{\phi_M}}{\partial\theta}, \theta, v)$
  $p += v$
**end for**

---

### A.13 CONVERGENCE RATE COMPARISON

We aim to learn optimisers with fast convergence speed. To evaluate the training speed of the learned MetaMD and to compare it with other methods, we measure the area under the curve (AUC) of the training loss (Rijn et al., 2015) where a small AUC value indicates an optimiser converges fast. Table 5 shows that the learned MetaMD has the fastest convergence speed in the CIFAR10 and ResNet18 setting while MetaMD ranks second in the High resolution setting in Table 6. To further illustrate the property of MetaMD, we compute the wall clock time of each optimisation iteration for each model in Table 7. Our model can still outperform other meta-learned precondition methods, such as MetaCur.

### A.14 FINE TUNING WITH LEARNED METAMD

We studied the transferability of the learned MetaMD by shifting dataset between the meta-train and meta-test stage. A different kind of domain shift is to deploy the learned MetaMD for fine-tuning the base model, which is different to the meta-train stage where the base model is learned from scratch in each each inner loop. We reuse the High resolution setting in Section 5.2, MetaMD is trained on all the meta-train datasets with the Algorithm 1 but deployed to fine-tune the ResNet18 pre-trained on ImageNet on each individual meta-test dataset in the meta-test stage. Compared with the hand-craft optimisers including, SGD-M and Adam, with fair hyperparameters selection, MetaMD has competitive performance in terms of test accuracy in Table 8.

Table 5: Area under the learning curve of the training loss for CIFAR10 and Resnet18. Comparison with various optimisers.

| Method | SGD-M | Adam | AdamW | KFAC | PowerSign | AddSign | Meta-SGD | MetaMD |
|---|---|---|---|---|---|---|---|---|
| CIFAR10 | 32.04 | 15.27 | 20.05 | 27.77 | 34.06 | 32.90 | 19.47 | **11.82** |

Table 6: Area under the learning curve of the training loss for high resolution task. Comparison with various optimisers.

| Method | SGD-M | Adam | PowerSign | AddSign | Meta-SGD | ARUBA | ARUBA ++ | MetaCur | MetaMD |
|---|---|---|---|---|---|---|---|---|---|
| Caltech | 106.80 | **69.77** | 108.23 | 111.86 | 176.59 | 409.97 | 327.83 | 736.83 | 90.20 |
| DTD | 434.23 | **94.35** | 138.44 | 133.88 | 153.68 | 509.23 | 379.49 | 469.52 | 132.84 |
| Flowers | 111.32 | **74.78** | 145.12 | 142.23 | 112.06 | 672.46 | 386.41 | 332.81 | 94.25 |
| Pubfig | 56.80 | **44.19** | 82.33 | 68.77 | 81.08 | 416.87 | 274.47 | 512.78 | 47.88 |
| Average Rank | 4.00 | 1.00 | 5.00 | 4.25 | 5.00 | 8.50 | 7.00 | 8.25 | 2.00 |

## A.15   NUMBER OF META-PARAMETERS

We compute the number of the meta-parameters for every meta-optimiser including Meta-SGD MetaCur and MetaMD in Table9. It can be noticed that MetaMD has a significantly small number of meta-parameters to learn compared with other meta-optimiser, which is one of the reasons that MetaMD suffers less from meta-overfitting.

## A.16   WEIGHT DECAY STRENGTH AS DOMAIN SHIFT

We simulate the domain shift by changing the weight decay strength to evaluate the transferability of the learned optimisers. During the meta-train stage, the optimisers are learned by setting the weight decay strength as zero, and we test the model on the meta-train dataset with various weight strengths including $\{0.0, 10^{-5}, 10^{-4}, 10^{-3}\}$. In Figure 9, we can notice that MetaMD is quite stable with the simulated domain shift compared with SGD and Adam and has the lowest AUC on all the settings.

Table 7: Wall clock time per iteration (mean and standard deviation) for High resolution task with ResNet18. Comparison with various optimisers.

| Method | SGD-M | Adam | PowerSign | AddSign | Meta-SGD | ARUBA | ARUBA ++ | MetaCur | MetaMD |
|---|---|---|---|---|---|---|---|---|---|
| Time | $118.21 \pm 2.37$ | $121.16 \pm 2.62$ | $121.30 \pm 2.38$ | $122.45 \pm 3.21$ | $120.29 \pm 3.32$ | $118.43 \pm 3.63$ | $118.38 \pm 3.62$ | $132.63 \pm 3.63$ | $131.12 \pm 3.62$ |

Table 8: Test Accuracy (%) and AUC on high resolution datasets by fine-tuning ImageNet pre-trained ResNet18. Comparison with various optimisers.

| Datasets | Caltech | DTD | Flowers | Pubfig | Avg Rank |
|---|---|---|---|---|---|
| SGD-M (Fine-tuning) | $77.81 \pm 0.73$ | $67.13 \pm 0.58$ | $91.56 \pm 0.54$ | $89.52 \pm 0.39$ | 2.5 |
| Adam (Fine-tuning) | $72.44 \pm 0.19$ | $68.46 \pm 0.77$ | $92.03 \pm 0.81$ | $90.12 \pm 0.28$ | 2.5 |
| MetaMD (Fine-tuning) | $77.93 \pm 0.44$ | $68.66 \pm 0.93$ | $91.67 \pm 0.75$ | $89.94 \pm 0.41$ | 1.5 |
| SGD-M (AUC) | 154.98 | 78.58 | 22.12 | 59.06 | 2.5 |
| Adam (AUC) | 70.64 | 42.39 | 26.86 | 59.62 | 2.0 |
| MetaMD (AUC) | 150.40 | 77.23 | 21.63 | 54.29 | 1.5 |

Table 9: Number of Meta-parameters for various Meta-optimisers

| Model | MetaSGD | MetaCur | MetaMD |
|---|---|---|---|
| | 11176512 | 2974717 | 37664 |

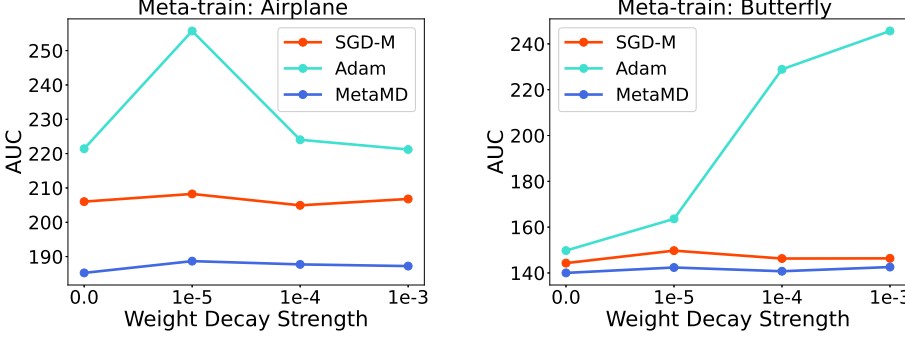

Figure 9: Area Under the Curve (AUC) comparison with various optimisers on different Weight Decay strength.

