# OpenReview forum: "MetaMD: Principled Optimiser Meta-Learning for Deep Learning"
_ICLR.cc/2023/Conference — Submitted to ICLR 2023_

### Official Review · Reviewer_f8bx · 2022-10-22

**Confidence:** 4
**Correctness:** 4
**Technical Novelty And Significance:** 3
**Empirical Novelty And Significance:** 2
**Recommendation:** 6

**Clarity, Quality, Novelty And Reproducibility:**

**Clarity + Quality:** The paper's writing is of high quality and I could understand all motivations, even though I am not an expert in this subfield. I found no major issues with the paper and consider it a solid contribution.

**Novelty + Originality:** While the method is not extremely novel, it is constructed motivated techniques from the standard literature, and is well-grounded based on both its conceptual and experimental contributions.



**Strength And Weaknesses:**

# Strengths
* Even though the proposed method has multiple steps, the paper is very well-written and conveys all such steps clearly, and makes itself accessible to even non-experts.
* The method appears to be quite natural, theoretically motivated, and utilizes well-known techniques from the literature (convex optimization, bilevel optimization, etc).
* The experimental results appear to be quite strong, with performance gains over numerous baselines within the few percentages, sometime within the tens of percentages on the "High Resolution Image Application" benchmark.

# Weaknesses
* While I am not an expert on the particular topic of Stochastic Mirror Descent, it appears that the method is only applicable to SGD-like optimizers, and momentum-based optimizers such as Adam are not represented here. If so, is there a way to extend the method to the momentum case? Momentum-based optimizers such as Adam are still the de-facto choice for applications such as reinforcement learning.
* The largest network used in this paper, ResNet-18, even if applied to 224 x 224 image sizes, is still quite small in the field of deep learning at the end of the day. While I am not explicitly requiring the authors to try the method on much higher compute budget experiments (e.g. training large Transformers, ResNet-101 on ImageNet-sized datasets), I do want to know if the method would still hold up, or if certain parts may need to be changed.
    * Answering this would greatly improve its impact over today's large-scale training, which in all honesty are probably the intended audience for such a work anyways, since the potential benefits over e.g. large language model training are far bigger than simply improving accuracies over MNIST or CIFAR10.

**Summary Of The Paper:**

In summary, this paper:
1. Proposes to attack the problem of learning gradient-based optimizers by considering the space of gradient updates which can be represented in the Stochastic Mirror Descent (SMD) framework.
2. In the SMD framework, the 1-step gradient updates are all parameterized by the expression of the Bregman Divergence. In the current paper's setting, quadratic costs $\theta^{T} M \theta$ with learnable matrix $M$ are considered.
3. Quite a few techniques are used to make learning $M$ feasible; they are primarily:
    * Expressing $M$ as block diagonal matrices to reduce the size, where each block corresponds to a layer in the neural network. Each block is also parameterized by a Kronecker factorization to further reduce computation.
    * Using an upper bound on convergence rate, as the actual objective to differentiate $M$ across.
    * Using implicit gradient descent to further reduce the explosive memory/computation blowups from auto-differentiating through a gradient update chain.
4. Experiments show solid results on toy quadratic functions, meta-training MNIST variants, and meta-testing on CIFAR10 + ResNet-18. Lastly, "larger" tasks were also experimented on with their own meta-train / meta-test, by increasing image resolutions over a distribution of datasets (Aircraft, Butterfly,...etc.) to 224x224.


**Summary Of The Review:**

The paper is a solid application of well-known optimization techniques in the literature, and has demonstrated this with both conceptual and experimental contributions. The main boost to the paper's score and impact, is if my question about applying the method over larger-scale experiments are answered.

---

> ### Author Response · Authors · 2022-11-17
> **Response to Reviewer f8bx**
>
> **Momentum:** Most existing work on mirror descent does not consider momentum terms, including the prior theoretical analysis that our approach is based on. So yes, this is a current limitation of our work. We are optimistic MetaMD could be upgraded to work with momentum when the underlying mirror descent theory and algorithms are correspondingly extended. However, we leave this to future work.
>
> **Scaling to larger networks:** Scaling up to large networks is a challenge for all learned optimisers, with most being tested on small LeNet scale CNNs.This is because of the compute cost of optimiser learning, and because larger networks make it easier to meta-overfit the optimiser. We are optimistic about the ability to scale MetaMD to larger networks because our empirical results on larger ResNets are already good, and because we have a principled way to include a regulariser to avoid meta-overfitting (Eq 8). However, due to the meta-train cost (one neural network training to completion for each meta-gradient step) and our limited GPU resources, we are unable to complete larger experiments within the feedback period. Finally, we emphasise that during deployment (i.e., meta-test) the computational footprint of our approach is only slightly higher than Adam, as now detailed in Table 7 of the appendix.

---

### Official Review · Reviewer_c6CM · 2022-10-25

**Confidence:** 2
**Correctness:** 3
**Technical Novelty And Significance:** 3
**Empirical Novelty And Significance:** 3
**Recommendation:** 8

**Clarity, Quality, Novelty And Reproducibility:**

First, I want to note that I am not an expert on learned optimizers.

## Novelty (W1)

Meta-learning the preconditioner has been considered before, e.g. [2], which is cited by the authors. Could the authors please comment on the differences between the proposed method and other approaches for learning the preconditioner?

## Technical (W2)

From the paper, it wasn't clear to me why in Eq. (8) you introduced $\theta_T^(i)$ in place of $\theta^*$? It seems like you could potentially try to 1 get the best possible estimate of $\theta^*$ for each of the meta-training tasks that you have, and then simply train $M$ with the sum of objective in Eq. (7) over the meta-training tasks. In this case, can you assume that $\theta^*$ does not depend on $M$?

Moreover, my understanding from Section 4.4 is that you essentially arrive at this formulation, where you take gradients of $B_{\phi_M}(\theta, \theta^1)$ where $\theta$ is the solution to the inner-problem that you obtain. Here, do you assume that $\theta^*$ does not depend on $M$?

Further, why did you separate the $1 / \lambda$ term from the $B_{\phi_M}$ term in Eq. (8)? Where does this exact expression come from?

## Experiments

I have a small question about the synthetic experiment: what is the optimal preconditioner here? My understanding is that for each given $Q$ the optimal preconditioner should be $Q^{-1}$? If so, how do you learn a preconditioner that is better than $I$ for random $Q$ matrices?

Similarly, could you provide some intuition into what preconditioner is learned more generally by your method? Is there some interpretable structure in the learned kronecker factors?

**Strength And Weaknesses:**

## Strengths

**S1**: The paper is clearly written and easy to follow, especially given the fairly technical subject. I enjoyed reading it.

**S2**: The proposed method is well-motivated theoretically, with connections to mirror descent.

**S3**: The authors derive a new bound on the generalization performance of their learned optimizer in Theorem 4.1 (I did not check the proof of this bound).

**S4**: Empirically, the method performs well on a range of tasks.

## Weaknesses

**W1**: While the authors present the method quite generally, as learning the Bregman  divergence for mirror descent, they restrict the class of these divergences so that the method is learning a preconditioner for SGD, which has been considered before.

**W2**: The motivation for the training objective in Eq. 8 is not immediately obvious to me.

I discuss the weaknesses in more detail below.

**Summary Of The Paper:**

The paper provides a method for meta-learning a preconditioner for gradient descent. The authors motivate the method from the mirror descent perspective, and use a convergence rate bound as a training objective. The authors parameterize the preconditioner in the K-FAC format [1]. Empirically, the method achieves good results across a range of problems.

**Summary Of The Review:**

Overall, this is a nice paper with good empirical results. I hope the authors can provide clarifications to my questions above.

## References

[1] Optimizing Neural Networks with Kronecker-factored Approximate Curvature
James Martens, Roger Grosse

[2] Meta-Learning with Warped Gradient Descent
Sebastian Flennerhag, Andrei A. Rusu, Razvan Pascanu, Francesco Visin, Hujun Yin, Raia Hadsell

---

> ### Author Response · Authors · 2022-11-17
> **Response to Reviewer c6CM**
>
> Thanks for your feedback.
>
> **W1.** Novelty, given that preconditioning matrix has been learned before?
> Yes. Preconditioners have been learned before, for example by KFAC and WarpGrad.  But there are important differences.
>
> **Re: KFAC.** We use similar parameterisation with KFAC [1]. However, instead of computing a new block diagonal matrix from the fisher information matrix in every update iteration, we learn a single block diagonal matrix in the meta-train stage and in the meta-test stage deploy it as a precondition matrix without any adaptation. This has stability benefits over KFAC because otherwise in a stochastic gradient regime the FIM has high variance w.r.t. mini-batch sampling, leading to our better performance, e.g., in Tab 2 (KFAC was excluded in other experiments because it failed to converge). It also shows how to accelerate substantially by amortising the cost of preconditioner learning  (we are much faster than KFAC in the deployment stage).
>
> **Re: WarpGrad.** MetaMD and WarpGrad are related in regard to learning preconditioners. But they differ in several important ways:
> 1.  Underlying formalism. WarpGrad is formalised based on Reinmann descent, while MetaMD is formalised based on Mirror descent.
> 2. Goal. WarpGrad is designed for few-shot problems and optimises for validation loss, MetaMD is aimed at the many shot regime and optimises for convergence rate.
> 3.  Parameterisation and transferability. MetaMD uses a compact KFAC style approximation to reduce the number of parameters to learn for better cross-task transferability, while WarpGrad learns a large number of parameters, which increases its risk of task-overfitting and reduces cross-task transferability compared to MetaMD. For example, when applied to ResNet18, in our high res experiment (Tab 3), MetaMD learns 37,664 meta-parameters, and WarpGrad learns 3,171,328 meta-parameters.
> 4.  MetaMD doesn’t modify the function that it is trying to optimise. WarpGrad modifies the function that it is optimising because warp layers remain in place during inference. WarpGrad essentially doubles the size of the final network during inference.
> 5.  Short horizon bias. WarpGrad requires a shorter inner loop due to memory requirements that grow with inner loop size, and therefore suffers from short horizon bias [A]. MetaMD uses implicit gradients to calculate meta-gradient memory-efficiently with a long inner loop, and thus does not suffer from short horizon bias. This leads to better performance during deployment.
>
> We have initiated an empirical comparison against WarpGrad. However the WarpGrad results so far are very poor (similar to MetaCurvature on RN18/High Res experiment). This may be due to the above issues, especially learning too many parameters and thus overfitting to the meta-train tasks. However, we are double checking our experiments to rule out other explanations such as poor tuning or convergence, and will update the paper accordingly if this is confirmed beyond a doubt within the rebuttal time-window.
>
> **W2.** Motivation and technical details about Equation 8
> Motivation: Motivated by learning optimisers with fast convergence, we derive our meta-objective from the bound of the stochastic convergence rate. Switching from dividing $ \lambda $ to adding $ 1/\lambda $  makes the hypergradient computation more stable in our case.
>
> **Details:** Sorry for the confusion. $ \theta_{\*} $ represents the optimal solution of training which is theoretically not achievable. We assume, based on the convergence guarantee associated with mirror descent, when T is large is enough $ \theta_{T} $ can approximate $ \theta_{\*} $ well. What we did in practice is the same as the reviewer described. Here we emphasise that reaching $ \theta_{\*} $ depends on M. In section 4.4, we reframe the optimisation problem when we need to compute the hypergradient with implicit gradient, because implicit gradient computation is not straightforwardly applicable to optimizer hyperparameters (as pointed out by [B]). In particular, the fixed point obtained by a mirror descent algorithm depends on the choice of Bregman divergence [C], and we use this concrete relationship to make the dependence on the choice of Bregman divergence appear in the objective of an equivalent optimisation problem, thus enabling use of the implicit gradient. The Bregman divergence is determined by the value of M, so fixed points depend on M.
>
>
> [A] Wu et al, Understanding Short-Horizon Bias in Stochastic Meta-Optimization, ICLR 2018.\
> [B] Lorraine J, Vicol P, Duvenaud D. Optimizing millions of hyperparameters by implicit differentiation. InInternational Conference on Artificial Intelligence and Statistics 2020.\
> [C] Azizan N, Lale S, Hassibi B. Stochastic mirror descent on overparameterized nonlinear models. IEEE Transactions on Neural Networks and Learning Systems. 2021.

---

> ### Author Response · Authors · 2022-11-17
> **Response to Reviewer c6CM**
>
> **Quadratic Problems**
>
> We agree that the optimal preconditioner for a specific quadratic problem is $Q^{-1}$. The identity matrix is not necessarily the optimal choice of preconditioner depending on the specific task family. The optimal preconditioner will depend on the specific distribution used to generate the Qs for each optimisation problem. Appendix A1 gives the details of the distribution we used for the experiment in Fig 1.

---

### Official Review · Reviewer_bxcN · 2022-11-02

**Confidence:** 2
**Correctness:** 4
**Technical Novelty And Significance:** 3
**Empirical Novelty And Significance:** 3
**Recommendation:** 5

**Clarity, Quality, Novelty And Reproducibility:**

See points above. The paper is well-written and clear, but Algorithm 1 could be more self-contained and detailed.

**Strength And Weaknesses:**

- The idea is well-motivated and the approach is promising. Adopting the Mirror Descent framework for meta-learning is something that has been discussed in the community but a practical yet expressive approach was still to be found, which I think this paper accomplishes. The form of the potential is not too constrained (although being block diagonal is somewhat restrictive) and the use of the IFT seems to fit well in this setting.
- The description of the method is mostly clear, but Algorithm 1 could have more details. Line 8 could be expanded to have the actual form of the hypergradient (following Eq 13) and what steps are used to compute it (Neumann series, as mentioned right before Section 5?). H seems to be an accumulator for the (hyper)gradients of M, and this could also be stated somewhere in Algorithm 1.
- Experimentally, the proposed method comfortably outperforms competing approaches in most tasks, but it would be very valuable to also consider commonly-adopted few-shot learning problems such as miniImageNet and MetaDataset. Some of the tasks used in the comparison seem less standard. Some of the considered methods also perform very poorly on the considered tasks but are known to perform well on miniImageNet (like MetaCurvature): it's unclear whether these methods are incompatible with such tasks for some reason, or if they were trained under less than optimal settings.
- A comparison against WarpGrad is expected but not given in the paper. This is an important evaluation since WarpGrad is somewhat motivated by MD and is more recent than most, if not all, of the considered competing methods.

**Summary Of The Paper:**

The paper approaches meta-learning from a mirror descent perspective, considering a setting where the potential of the Bregman divergence is meta-learned. It proposes a method that can be efficiently implemented by restricting the form of the potential and using the IFT to be practical. Experiments show improved performance compared to competing methods.

**Summary Of The Review:**

The proposed method strengthens the idea that the MD framework can be successfully adopted for meta-learning, and while a compromise is made in terms of expressivity (the form of the potential is still somewhat restrictive) to achieve reasonable computational costs, the experiments show improvements over competing methods nonetheless.

The main weaknesses, in my opinion, are the lack of results on more standard tasks such as miniImageNet and MetaDataset, and a missing comparison against WarpGrad. These would be extremely valuable and would enable a more rigorous and complete evaluation of MetaMD.

------------

I believe reviewer Ced8 raises key concerns which I had overlooked in my initial review. The authors' response does not fully address these points, especially regarding the suboptimal performance of baseline methods due to the adoption of lightweight HPO instead of using hyperparameter values recommended by the corresponding papers. This could result in an unfair advantage to MetaMD due to a possible increased robustness to hyperparameter settings -- an advantage that might only manifest when a low budget HPO is deployed which is effectively choosing poor hyperparameter values. In light of this, I have decided to decrease my score.

---

> ### Author Response · Authors · 2022-11-17
> **Response to Reviewer bxcN**
>
> Thanks for your time and helpful feedback.
>
> **C1:** Description of Algorithm 1
> Following the reviewer’s suggestion, we have added details into Algorithm 1 to better explain the computation of H. Specifically, in Appendix A12, we add Algorithm 2 and expand Eq. 13 with the Neumann series to provide a detailed explanation of the inverse Hessian matrix computation.
>
> **C2:** Comparison on FSL
> We agree that Few-shot learning is an interesting research area, and we repurposed some FSL-oriented learned optimisers (MetaSGD, metaCurvature) for baselines in our evaluation, but the main focus of this paper is to learn optimisers for the many-shot setting (as do KFAC, Aruba, L2O). The reasons that methods such as MetaCurvature perform well on few-shot miniImageNet yet underperform on our benchmark are essentially that it failed to scale to the larger networks and many-shot regime. Specifically: (1) MetaCurvature was originally demonstrated only on small CNNs and low res images, and in this regime it also performs well in our experiments (Tab 1). However, in the larger ResNet CNN/high res image regime, it has to learn many more parameters and it overfits to the source tasks and performs poorly on the target tasks (Tab 3). (2) In order to tractably apply MetaCurvature to our problem we had to apply it with a short inner loop during meta-training. This causes it to suffer from the short horizon bias problem [A]. Our MetaMD doesn’t suffer from this problem, because we can efficiently meta-train with long inner loops due to use of implicit gradients.
> Re: “Lack of results on more standard tasks”. Please note that our tasks are quite standard CNN learning tasks for the many-shot regime.
>
> **C3:** Comparison to WarpGrad
> We have implemented and run an empirical comparison against WarpGrad. However the WarpGrad results so far are very poor (similar to MetaCurvature on RN18/High Res experiment). This is probably due to WarpGrad learning too many meta-parameters compared to MetaMD and thus overfitting to the meta-train tasks. (MetaMD learns 37,664 meta-parameters, and WarpGrad learns 3,171,328 meta-parameters). However, we are double checking our experiments to rule out other explanations such as poor tuning or convergence, and will update the paper accordingly if this is confirmed beyond a doubt within the rebuttal time-window.
>
> [A] Wu Y, Ren M, Liao R, Grosse R. Understanding short-horizon bias in stochastic meta-optimization. arXiv preprint arXiv:1803.02021. 2018 Mar 6.

---

> ### Author Response · Authors · 2022-12-13
> **Important response to big score decrease**
>
> We believe that decreasing the score from 8 to 5 at the last minute is a very large change, which needs more detailed discussion. It will not be acceptable for us if 3 points are lost without serious consideration. **We would like the reviewer to raise the new official comment so that we can get the notification and give immediate responses in the further rounds instead of editing the previous messages which does not trigger notification to authors.**
>
> We think the reviewer take the wrong idea about the complaint from reviewer Ced8. The concern is from the results in Table 3. As we clarified that Reviewer Ced8 assume that we conducted our experiments on Fine-tuning setting which is not what we did. Our study focuses on a learning optimiser that trains the model from scratch. We have justified in our response to Ced8 that “The reviewer suggests that > 75% accuracy to be credible for Caltech-256, but this is in line with what is achievable by fine-tuning a pre-trained model (https://arxiv.org/pdf/2111.04578v1.pdf, NeurIPS’21), and is certainly not a reasonable target for training ResNet18 from scratch. Meanwhile, the closer 47% target mentioned by the cited GitHub link is also not for ResNet18 and uses data augmentation. Moreover, there is no standard train/test split for Caltech-256, so results are generally not comparable across papers.” In addition, we have emphasised that we did a fair comparison with enough hyperparameter tuning for this setting. Still, we are waiting for the response from Ced8 and we believe most of the questions from Ced8 have been addressed.
>
> We know that in the final stage your work is overwhelming and our response to reviewer Ced8 is massive, but we kindly suggest the reviewer read our response more carefully with our updated experiments required by Ced8 in the appendix. We are happy to answer more questions and more feedback is expected. We kindly suggest the reviewer read everything carefully and reconsider the scores.

---

### Official Review · Reviewer_Ced8 · 2022-11-04

**Confidence:** 4
**Correctness:** 2
**Technical Novelty And Significance:** 3
**Empirical Novelty And Significance:** 2
**Recommendation:** 3

**Clarity, Quality, Novelty And Reproducibility:**

I reiterate that the paper is well written with a good motivation and presentation of the method. Further, the meta-optimization principle seems to work in practice. However, as I explain above, I am more concerned with the execution of the experiments, which are not justifying well that there is a benefit of the meta-optimization in MetaMD: There is no clear quantification of the transferability of the meta-learned divergence (Rotated digits and DiverseDigits seems an artificial task with merely no difference across data distribution). Technically, while the block-diagonal construction of the quadratic cost for the Bregman divergence is interesting, there is limited novelty since MetaMD makes a direct use of the implicit differentiation framework.




**Details Of Ethics Concerns:**

-

**Strength And Weaknesses:**

Strength : From a formal point of view, I feel that this work is clear and well written : understanding is straightforward, the authors motivates well the proposal, which is further discussed in comparison with the relevant meta-optimization literature. The theoretical and experimental results are also clear without ambiguity.

However, I have several concerns regarding the applicability of the method and the execution of the experimental section that I list below in order of importance:

 - I) In my opinion, an important question is how transferable is the learnt divergence when applied to new optimization problems, as it determines the genuine applicability of the method and the benefit of the expensive meta-training procedure. It seems that this is very partially answered in the experimental sections with mixed results: In the DiverseDigits experiment, the training loss function seems to plateau at a slightly higher level than two off-the-shelf optimizers without meta-adaptation: RMSProp and Adam, which calls into question the transferability of $B_{\phi}$ for datasets that are most dissimilar to the meta-training curriculum. In order to form an idea of how general is the learnt meta-optimizer, It would be beneficial to the paper to provide a quantification of the method transferability when controlling tasks dissimilarities (by varying datasets or the loss function for instance).

- II) Another point that weakens the paper argument is that there is no empirical results regarding the acceleration proposed by this method, whereas it is one claim of the abstract. Apart from the iteration count in the toy dataset, no quantification of the acceleration induced by metaMD is provided. Worse, the different optimization baselines are not controlled for computational budget (such as total number of flops or time for reaching stopping criterion for instance). The meta-training procedure is compute-intensive and the argument that it constitutes a "one-off cost that can be amortized across different meta-test problems of interest" would hold only if transferability (point ii) is valid which is hardly decidable from the current set of experiments.

 - III) Some of the performance metrics seems surprisingly low compared to what is usually reported in the literature. For instance the accuracy regime of ResNet-18 for Caltech and Flowers dataset is lower than 30% for all baselines. By inspecting more closely the training curves in Appendix (figure 6) It seems that this low test performance mainly comes from substantial overfitting. To the author point, the benefit of MetaMD is to provide an implicit regularization of the optimization problem (since meta-training is performed on the other datasets). Since classification test accuracy is the only selected metric, the unusual overfitting regime in which the tested optimization scheme are compared seems to artificially favor MetaMD.

 - iv) One explicit motivation of the paper for meta-optimizing a stochastic mirror descent scheme (SMD) is to retain the well-established theoretical guarantees offered by SMD method. However, the authors propose a convergence rate in the convex setting, which is fairly restrictive and ill-aligned with the experiments over non-convex problems carried in the paper. There exists a rich literature on convergence of SMD in the non-convex case and I feel that such theoretical results provided are uninformative.


**Summary Of The Paper:**

This work proposes a meta-optimised mirror descent scheme for large-scale optimization problems, based on tuning of a parametric Bregman divergence on a set of curriculum meta-training task. The expected benefits of this meta-learning procedure is to produce an optimiser with accelerated convergence that is transferable to similar optimization problems, while retaining theoretically provable convergence guarantees. The authors provide a theoretical discussion regarding Stochastic mirror descent as well as an optimization framework leveraging the Implicit Function Theorem for meta-optimizing the Bregman divergence used in the mirror descent scheme. The authors further describe a particular construction of the Bregman divergence based on block-diagonal Kroenecker factorization of a quadratic form, aiming at finding a trade-off between computational complexity of the inner optimization problem and meta-adaptability, as well a convergence result of this approach in the convex case. The paper provide experiments showing the doability and the potential benefit of the method (in terms of test accuracy) on a set of synthetic problems / MLP and Convolutional NN training on MNIST-like datasets classification/ ResNet-18 on higher resolution images dataset classification.

**Summary Of The Review:**

Overall, while I personally find this line of work very interesting and would like proposals such as MetaMD to come through, I feel that the current experimental section has too many weaknesses which makes this submission not ready for publication. I am willing to discuss more specifically with the authors the points (I-IV) raised above and to increase my score should they provide more compelling experimental evidence, in priority regarding transferability and acceleration.

---

> ### Author Response · Authors · 2022-11-15
> **Preliminary review clarification questions**
>
> Thanks for your feedback. We will answer your questions in detail soon, providing details on acceleration, etc. But first we would like to ask for some clarification on some of your concerns (weakness I, IV) to make sure we answer appropriately.
>
> **W-I. Transferability:**
>
> To recap, our existing experiments have four empirical evaluations of transferability.
> 1. RMNIST: Transfer across rotations (leave one rotation out)
> 2. DiverseDigits: Transfer across datasets with different character sets and styles (leave one dataset out)
> 3. CIFAR10: Meta-Train: DiverseDigits+STL10 => Meta-Test: CIFAR-10
> 4. High res: Meta-Train: (Aircraft, Butterfly, Pets) => Meta-Test: (Caltech, DTD, Flowers, Pubfig)
>
> We viewed empirical success above, especially on (3 and 4), as a clear demonstration of transferability across substantially different optimisation problems.
> We agree that learned optimiser transferability is a challenge and do not claim it is completely solved. However, we viewed the results above as showing that our MetaMD solves it better than other meta-learned competitors that aim to transfer knowledge across tasks such as L2O/MetaCurvature/MetaSGD/etc. Empirically, we can also often beat non-meta learned Adam and SGD-M on these novel problems, which we also viewed as a successful demonstration of cross-task knowledge transfer.
>
> We are not clear on what kind of experiment you imagine would more decisively demonstrate transferability. It would be appreciated if you can elaborate on why you think result (3 and 4) do not successfully demonstrate transferability, and what kind of experiments you think would successfully demonstrate it.
>
> Your review mentioned it would be beneficial to quantify transferability while controlling task similarity. We agree this would be interesting, however evaluating task similarity is itself an open question. We considered measuring the distance between our meta-train and meta-test tasks in terms of task2vec [A] distance, in order to compare this with downstream performance. But this is confounded by the different underlying difficulty of the downstream tasks, so there is no straightforward way to quantitatively correlate task similarity and performance.
>
> Your review also mentioned being interested in transferability across different loss functions. We can do this experiment if it is considered particularly interesting, but it wasn’t obvious to us that this is a stronger demonstration of transferability than transfer across datasets that we already evaluated. Since there is not a straightforward way to measure similarity between loss functions, this also doesn’t seem to solve the previous point about quantifying transferability as a function of task similarity.
>
> **W-IV:**
>
>  Your review mentioned a rich literature on SMD convergence in the non-convex case. We are not aware of any literature in this area that can provide non-asymptotic convergence rates when training neural networks. The papers that we have found consider classes of non-convex functions that do not seem to include neural networks. Could you provide some more specific pointers?
>
> [A] Achille A, Lam M, Tewari R, Ravichandran A, Maji S, Fowlkes CC, Soatto S, Perona P. Task2vec: Task embedding for meta-learning. InProceedings of the IEEE/CVF international conference on computer vision 2019 (pp. 6430-6439).

---

> > ### Comment · Reviewer_Ced8 · 2022-11-16
> > **Response to authors**
> >
> > I appreciate this anticipative reply and detail more precisely below the weaknesses that I pointed out in my initial review:
> >
> > - Two types of arguments regarding  your experiments lead me to think that transferability of MetaMD is not correctly assessed. While I appreciate that you ran several different experiments to support the claim that MetaMD learnt divergence can provide a beneficial regularisation signal to meta-test optimization problems, I think that it is not compared in a sufficiently fair setting (point III) for the following reasons:
> >
> >  	- RMNIST/Transfer across rotations: Unless I missed it in the supplementary, I found no precise description of the architecture that you refer as a “3-layer MLP”.  However, a rapid inspection of results similar to your setting at http://yann.lecun.com/exdb/mnist/ shows that we can achieve a lower error rate for baselines (See section “Neural Nets”) This calls into question whether reported accuracies reflect true optimality in your optimisation pipeline and the marginal gain that MetaMD brings in terms of accuracy for this experiment.
> >
> > 	- DiverseDigits. The arguably two most different datasets from the others (FashionMNIST & SVHN) are showing no improvement over baseline optimisers (Adam and SGD respectively (Table 1 & figure 3.3) when MetaMD is deployed. This fact directly undermines the transferability of MetaMD.
> >
> > 	- CIFAR10 and High res: For CIFAR10, results for SGD-M and Adam are again a few point of percentage below what can be found for comparable experimental settings https://paperswithcode.com/sota/stochastic-optimization-on-cifar-10-resnet-18, and I suspect the difference to come from data augmentation. I found no reference to it in the supplementary. Hence, I would be interested in checking if MetaMD holds its regularisation benefit in terms of accuracy in the context of data augmentation for baselines.  For the other High resolution experiment, results baselines are much less aligned with a rapid search on comparable results (I could find Chttps://par.nsf.gov/servlets/purl/10156604 for Caltech-256 which reports a 76.77% accuracy with SGD-Momentum compared to 26.95% in your experiment, I also found https://github.com/nickbiso/Keras-Caltech-256 which reports 47% accuracy with Adam.). I’m wondering what could cause such a difference in reported results. This reinforce my concern that this set of experiments might be comparing fairly unoptimal results which are much less informative of the transferability of MetaMD.
> >
> >  - Regarding assessing transferability in point I ) : Since ultimately, the goal of MetaMD would be to be deployed to new optimization problems, it seems important to provide at least proxies for understanding the robustness of the method to variation of the problem. I agree that evaluating task similarity can be arduous for image classification. However as I mentioned, data is a single aspect of empirical optimization problem and varying the loss function is an equally valid aspect, that is of particular interest for the community of meta-learning and lifelong learning. For instance, varying parametrically the loss landscape is much more tractable study (with a regularisation term or with synthetic data generation for instance, which would provide an explicit metric to quantify transferability). More generally, it would be beneficial to add experiment regarding other type of optimization problems such as regression rather than solely image classification.
> >
> > - Regarding point IV), Indeed, my main concern is that result 4.1 refers to convex functions which does not hold for considered problems in the experimental section. For non-smooth function more similar to current NN see for instance [1] and the notion of $(\delta,\epsilon)$-stationarity.
> >
> > [1] Complexity of Finding Stationary Points of Nonsmooth Nonconvex Functions, Jingzhao Zhang, Hongzhou Lin, Stefanie Jegelka, Ali Jadbabaie, Suvrit Sra

---

> ### Author Response · Authors · 2022-11-18
> **Response to Reviewer Ced8**
>
> Thanks for the detailed feedback. We provide detailed answers as follows.
>
> **Q1. Quantification of transferability.**
>
> The reviewer suggested quantifying transferability with respect to task similarity by varying the loss function between meta-train and meta-test. We agree this is a good and interesting idea that evaluates a different dimension of transferability than the cross-dataset transferability we focused on previously. To this end, we took our model trained from the High Res experiment (where l2 regularisation was disabled), and deployed it on some of the same datasets where we varied the l2 regularisation strength.  Since these are datasets available in meta-training, it’s expected that MetaMD outperforms Adam and SGD-M at weight decay=0, and the question is whether it continues to do so as the (meta) train-test task similarity is reduced by increasing weight decay.
>
> The results in Appendix A.16, show that when gradually reducing task similarity by way of introducing increasing weight decay strength, the performance of MetaMD in terms of training loss AUC remains quite robust as weight decay strength increases. In particular, it continues to outperform SGD-M and Adam across the whole operating range. This varying of regularisation is perhaps a lesser demand on transferability robustness compared to varying datasets as in Table 3. But it shows that MetaMD performance is robust enough to beat the baselines across the range evaluated.
>
> **Q2. Empirical results regarding acceleration.**
>
> Thanks for this good point, and sorry for the oversight. To quantify acceleration, we now report the area under the training loss curve (AULC, [C]) as a measure of optimisation quality and convergence rate, and wall-clock time per iteration in milliseconds. The convergence results (below and in Tables 5, 6 of the revised paper), show that MetaMD has the smallest AULC on CIFAR-10/ResNet18, and ranks second for AULC in the high-resolution task.  The clock-time results (below and in table 7 of the revised paper) show that all methods have a fairly similar cost, with MetaMD not being noticeably more costly than competitors.
>
> [A] Lorraine J, Vicol P, Duvenaud D. Optimizing millions of hyperparameters by implicit differentiation. AISTATS 2020.\
> [B] Azizan N, Lale S, Hassibi B. Stochastic mirror descent on overparameterized nonlinear models. IEEE Transactions on Neural Networks and Learning Systems. 2021.\
> [C] van Rijn JN, et al. Fast algorithm selection using learning curves. IDA 2015.\
> [D] Balkan M, Khodak M, Talwalkar A. Provable Guarantees for Gradient-Based Meta-Learning. ICML 2019.\
> [E] Reddi S, Kale S, Kumar S. On the Convergence of Adam and Beyond. ICLR 2018.\
> [F] Zhuang J, et al. AdaBelief Optimizer: Adapting Stepsizes by the Belief in Observed Gradients. NeurIPS 2020.\
> [G] Heo B, et al. AdamP: Slowing Down the Slowdown for Momentum Optimizers on Scale-invariant Weights. ICLR 2021.

---

> ### Author Response · Authors · 2022-11-18
> **Response to Reviewer Ced8**
>
> **Q3. Evaluation of Transferability and Explanation for low accuracy on high res datasets?**
>
> Initial Remarks: There were four cross-dataset experiments in Tab 1, Tab 2 and Tab 3, which we considered to demonstrate the transferability of MetaMD. We understand from our previous discussion that the reviewer’s main concern, aside from those already addressed in Q1 and Q2 above, is that the absolute numbers reported were lower than expected, especially in the experiments that were a more substantial transferability test (Tab 2, Tab 3), and this caused the reviewer to doubt the collective conclusion from all those results.
>
> We emphasise that our experiments were scrupulously optimised for fairness, with a lightweight validation set based HPO used consistently for all methods during meta-testing (see Appendix A7) and no manual hyper-parameter tuning involved. This is in contrast to the much work in the literature which is oriented around beating benchmarks and where heavy hyperparameter tuning is involved, often including manual hyperparameter tuning on the test-set. While heavier hyperparameter tuning (more HPO iterations and more data augmentation) would no doubt improve results for all methods to some extent, we disagree with the reviewer that this invalidates the current conclusion (MetaMD outperforming baselines in Tab 2 + Tab 3), and we do not believe it would change the overall conclusion. If we wanted to manually tune MetaMD hyper-parameters to match any other given manually-tuned benchmark number we could certainly do so, but we believe that this is scientifically less interesting than comparing a suite of baselines in a carefully controlled common HPO setting.
>
> With particular regard to the most substantive experiment in Tab 3, we view the results as showing: (1) That MetaMD outperforms the other meta-learned optimisers applied in a cross-dataset scenario. This is because these other methods fail to transfer across datasets due to overfitting to the source datasets and thus performing poorly on the target datasets. The baselines are failing the cross-dataset transferability test that the reviewer is interested in, while MetaMD passes it.
> (2) That MetaMD outperforms the non-meta-learned methods (SGD-M, Adam). This shows that MetaMD does successfully leverage source data to help improve the performance on the target dataset beyond what can be achieved by standard methods on the target dataset alone. We view this as being a clear demonstration of useful transferability that the reviewer is interested in seeing.
>
> The remaining point of contention seems to be that the absolute numbers in Tab 3 are lower than the reviewer expects, which makes them doubt the whole setup. We are quite sure that these numbers are reasonable, and we attribute the difference between our results and the reviewer’s expectation to the fact that many papers using these datasets applied (ImageNet pre-trained) fine-tuning, rather than train-from-scratch setting. The reviewer suggests that > 75% accuracy to be credible for Caltech-256, but this is in line with what is achievable by fine-tuning a pre-trained model (https://arxiv.org/pdf/2111.04578v1.pdf, NeurIPS’21), and is certainly not a reasonable target for training ResNet18 from scratch. Meanwhile, the closer 47% target mentioned by the cited github link is also not for ResNet18 and uses data augmentation. Moreover, there is no standard train/test split for Caltech-256, so results are generally not comparable across papers.
>
> To clarify this issue, we also repeat the high resolution experiment from Tab 3 using our same meta-learned optimizer, but deploy it to fine-tune an ImageNet pre-trained ResNet18 meta-test only. Note that in this experiment MetaMD now suffers two kinds of task shift simultaneously – both a cross-dataset transfer, and a regime transfer from train-from-scratch to fine-tune regime. The results in Tab 8 of the Appendix show that: (i) Fine-tuning the results are much higher (>75% accuracy on Caltech). This is as expected, and confirms that the explanation for comparatively low performance in Tab 3 is indeed the train-from-scratch condition. (ii) The performance of all models is much closer now (because SGD and Adam suffer much less from overfitting in the fine-tuning regime). (iii) Despite the rather huge task-transfer, MetaMD still performs reasonably well compared to Adam and SGD, achieving the higher average rank in both test accuracy, and convergence rate as measured by AUC.
>
> In summary, we believe our numbers are representative of reasonable performance for training  ResNet18 from scratch on these tasks without augmentation, that they are reasonable evidence to conclude that MetaMD demonstrates (i) improved transferability over other meta-learned competitors, and (ii) a useful margin of improvement over non-meta-learned baselines.

---

> ### Author Response · Authors · 2022-11-18
> **Response to Reviewer Ced8**
>
> **Q4. Convergence result only in the convex setting.**
>
> Thanks for pointing out the work by Zhang et al. which we cite in the revised version. Based on our understanding, the cited reference does not address mirror descent. We would actually be very interested to know about convergence results for non-convex SMD because, if these exist, we believe we could leverage them quite easily to improve our Theorem 4.1 for MetaMD in the non-convex case. Nevertheless, we emphasise that it is a very well-established route within the community to demonstrate theoretical support for an algorithm in the convex case and then evaluate it empirically also in the non-convex case [D,E,F,G], so this is not a major limitation of our work.
>
> **Q5. How novel is MetaMD’s “Direct use of implicit differentiation”?**
>
> We disagree that this is a trivial application. Conventional applications of the implicit gradient in meta-learning can only be applied to hyperparameters that explicitly appear in the training objective function. This means that existing implicit gradient methods can not be used to learn optimiser hyperparameters. The scope and limitation section of the authoritative implicit gradient study [A] says, “our approach is not straightforwardly applicable to optimizer hyperparameters”. One of our contributions is to reframe the optimisation problem by utilising the Mirror Descent property that mirror descent finds the fixed point closest to the initial guess of the network parameters as measured by the corresponding Bregman divergence [B]. This innovative transformation allows us to bypass the limitation stated by [A].
>
> [A] Lorraine J, Vicol P, Duvenaud D. Optimizing millions of hyperparameters by implicit differentiation. AISTATS 2020.\
> [B] Azizan N, Lale S, Hassibi B. Stochastic mirror descent on overparameterized nonlinear models. IEEE Transactions on Neural Networks and Learning Systems. 2021.\
> [C] van Rijn JN, et al. Fast algorithm selection using learning curves. IDA 2015.\
> [D] Balkan M, Khodak M, Talwalkar A. Provable Guarantees for Gradient-Based Meta-Learning. ICML 2019.\
> [E] Reddi S, Kale S, Kumar S. On the Convergence of Adam and Beyond. ICLR 2018.\
> [F] Zhuang J, et al. AdaBelief Optimizer: Adapting Stepsizes by the Belief in Observed Gradients. NeurIPS 2020.\
> [G] Heo B, et al. AdamP: Slowing Down the Slowdown for Momentum Optimizers on Scale-invariant Weights. ICLR 2021.

---

> ### Author Response · Authors · 2022-11-28
> **Response to Reviewer Ced8**
>
> Thanks again for your feedback.\
> We believe our last response answers all your questions.
> Please let us know if you have any further queries, or if not please consider raising your score.

---

### Author Response · Authors · 2022-11-18
**General Response from Authors**

Dear AC and reviewers,

We thank the reviewers for the time they have put into reviewing the paper and for their helpful feedback. The constructive suggestions have helped us to improve our paper further.  We appreciate that reviewers find the problem we study of significant importance and our experiments extensive. In general, the reviewers all find our novel approach of learning an optimiser with fast convergence under the Mirror Descent framework concrete. We have addressed all the reviewers’ concerns by changing some of the languages, modifying/adding figures, and including some analysis sections.

Here is a summary of our updates:
- **[Clarification of writing in Appendix 12]** We have further explained the details that reviewers are interested in or feel unclear about. The corresponding contents have been added or modified in our revised version. For example, we give a detailed explanation of the implicit gradient computation.
- **[Adding convergence rate quantification section in Appendix 13]** In the initial submission, we already generate a learning curve to compare the convergence rate of different optimisers. Thanks to the reviewers’ suggestions, we add a section quantifying the convergence rate of all the methods with the area under the training loss curve to give a more efficient illustration.
- **[Analysis of the model complexity in Appendix 15]** We analyse the reason for the meta-overfitting by comparing the numbers of learnable parameters in different optimisers.
- **[Analysis of Transferability with simulated domain shift in Appendix 16]** We further evaluate the meta-generalisation of the learned optimiser with controllable domain shift by varying the strength of the weight decay and compare the behaviour of our model with other optimisers such as SGD and Adam with noticeable better convergence speed.
In light of this, we would appreciate it if the reviewers would reconsider their final scores. We are happy to answer any other additional questions and provide more information.

Regards,

Authors

---

### Decision · Program_Chairs · 2023-01-20

**Decision:**

Reject

**Justification For Why Not Higher Score:**

There are criticisms from the reviewers based on the empirical protocols.

**Justification For Why Not Lower Score:**

There are reviewers championing the paper.  An expert on optimizer tuning seemed to feel that the empirical setup was well thought out and justified.  All reviewers seemed to appreciate the other merits of the paper (besides the raised experimental concerns).

**Metareview: Summary, Strengths And Weaknesses:**

This paper develops a methodology to meta-learn a mirror-descent based optimizer for deep learning models, which they call Meta Mirror Descent or MetaMD.  The optimization algorithm depends on a flexible pre-conditioner, that is applied in a manner similar (but motivated differently) to the Kronecker-factored optimization algorithm (KFAC).  The authors present theoretical justification of their approach in the form of convergence rate proofs.  In a variety of experiments, the authors demonstrate that their optimizer converges to better solutions than other optimizers such as KFAC, Adam-variants, and other baselines.  The authors used a Bayesian optimization setup to tune the hyperparameters of each algorithm, controlling for the same hyperparameter tuning methodology and effort across algorithms.

Initially, the review scores were quite positive, with (8, 8, 6, 3).  The reviewers in general found the work well motivated, well written, methodologically sound and interesting, and experiments convincing.  The main weakness cited was regarding the empirical evaluation setup and specifically to the execution of the experiments.  In particular, it seems a key criticism was that baselines were lower than in other papers because the authors used hyperparameters from their own hyperparameter tuning runs, rather than those taken from existing papers.  After the discussion period, one of the reviewers changed their score from 8 to 5, after reading the other reviews and author responses, citing this concern about empirical evaluation.  Thus after the review period the average review score was borderline reject (8, 5, 6, 3).

Given that all reviewers seemed to appreciate the merits of the paper, and the main point of contention was the experimental setup, I solicited the opinion of an expert on optimizer tuning practices.  Their opinion was that the practices followed by the authors, i.e. putting all methods on a level playing field and using the same hyperparameter tuning method (Bayesian optimization) with the same budget seemed well justified.  It demonstrates that there is a reasonable regime in which this approach is better (though perhaps exhaustively tuning or using expert tuning could do better).

This paper was further discussed with the SAC and the program chairs, weighing the strengths and weaknesses.  In that discussion, novelty and, in particular relation to existing work in the writing, was considered to be a weakness.  Ultimately, the decision was to reject the paper, in part because the remaining reviewer giving an 8 unfortunately indicated low confidence tipping the weighted balance to reject.

**Summary Of Ac-Reviewer Meeting:**

This paper became borderline on Dec 12, when one reviewer changed their score from 8 to 5.  Therefore, there wasn't time to initiate a meeting. Instead, I found an expert on the subject of contention and had them read the paper and reviews and provide an opinion.